# Pulmonary pericytes regulate lung morphogenesis

Katsuhiro Kato[1], Rodrigo Diéguez-Hurtado[1], Do Young Park[2,3], Seon Pyo Hong[2,3], Sakiko Kato-Azuma[1], Susanne Adams[1], Martin Stehling[4], Britta Trappmann[5], Jeffrey L. Wrana[6,7], Gou Young Koh [2,3] & Ralf H. Adams [1]

Blood vessels are essential for blood circulation but also control organ growth, homeostasis, and regeneration, which has been attributed to the release of paracrine signals by endothelial cells. Endothelial tubules are associated with specialised mesenchymal cells, termed pericytes, which help to maintain vessel wall integrity. Here we identify pericytes as regulators of epithelial and endothelial morphogenesis in postnatal lung. Mice lacking expression of the Hippo pathway components YAP and TAZ in pericytes show defective alveologenesis. Mutant pericytes are present in normal numbers but display strongly reduced expression of hepatocyte growth factor leading to impaired activation of the c-Met receptor, which is expressed by alveolar epithelial cells. YAP and TAZ are also required for expression of angiopoietin-1 by pulmonary pericytes, which also controls hepatocyte growth factor expression and thereby alveologenesis in an autocrine fashion. These findings establish that pericytes have important, organ-specific signalling properties and coordinate the behavior of epithelial and vascular cells during lung morphogenesis.

[1] Department of Tissue Morphogenesis, Max Planck Institute for Molecular Biomedicine, and University of Münster, Faculty of Medicine, D-48149 Münster, Germany. [2] Center for Vascular Research, Institute of Basic Science (IBS), Daejeon 34141, Republic of Korea. [3] Graduate School of Medical Science and Engineering, and Department of Biological Sciences, Korea Advanced Institute of Science and Technology (KAIST), Daejeon 34141, Korea. [4] Electron Microscopy and Flow Cytometry Units, Max Planck Institute for Molecular Biomedicine, D-48149 Münster, Germany. [5] Bioactive Materials Laboratory, Max Planck Institute for Molecular Biomedicine, D-48149 Münster, Germany. [6] Centre for Systems Biology, Lunenfeld Tanenbaum Research Institute, Mount Sinai Hospital, Toronto, ON M5G 1X5, Canada. [7] Department of Molecular Genetics, University of Toronto, Toronto, ON M5S 1A8, Canada. Correspondence and requests for materials should be addressed to K.K. (email: katsuhiro.kato@mpi-muenster.mpg.de) or to R.H.A. (email: ralf.adams@mpi-muenster.mpg.de)

Blood vessels form an extensive network of highly branched endothelial tubules that are covered by specialized supporting cells, pericytes, and vascular smooth muscle cells, surrounded by an extensive amount of extracellular matrix. Pericytes and capillary endothelial cells (ECs) contact each other and share the vascular basement membrane[1]. In addition to the delivery of circulating cells, oxygen and nutrients, blood vessels also provide instructive signals controlling organogenesis,

homeostasis, and regeneration[2–4]. The pulmonary vasculature has characteristic physiology and functionality, which centers around a complex alveolar gas exchange unit composed of a thin alveolar epithelium and a closely associated capillary plexus[5,6]. Previous work has established that proper growth and function of the vascular endothelium is indispensable for lung development, homeostasis, and regeneration[2,7–9]. By contrast, pulmonary pericytes have been mostly implicated in lung fibrosis[10,11] and

pulmonary hypertension[12], whereas their physiological function remains largely uncharacterized.

Alveologenesis, which is mainly a postnatal event between postnatal day (P) 5 and 30 in mice achieved through secondary septation subdividing the alveolar sac, is a highly integrated process that involves cooperative interactions between alveolar type 1 (AT1) and type 2 (AT2) epithelial cells, ECs, and a number of different mesenchymal cell types[5,6]. The disruption of this coordinated process has been implicated in neonatal diseases such as bronchopulmonary dysplasia (BPD).

Hippo signalling is a potent regulator of development, differentiation, and tissue homeostasis. The transcriptional co-activator Yes-associated protein 1 (Yap1) and WW domain containing transcription regulator 1 (WWTR1 or Taz), which binds to TEA domain (TEAD) proteins to form an active transcriptional complex controlling gene expression, are crucial for these functions. Yap1/Taz are phosphorylated by the complex of large tumor suppressor homolog 1/2 (Lats1/2) kinase and MOB kinase activator 1 (MOB1), which are activated by serine/threonine kinase 3 (Stk3/Mst2) and 4 (Stk4/Mst1) and the Salvador Family WW Domain Containing Protein 1 (Sav1) complex, leading to the inactivation of Yap1/Taz through exclusion from the cell nucleus and the promotion of proteolytic degradation[13]. Global genetic deletion of *Wwtr1* results in development of multiple renal cysts and pulmonary emphysematous changes[14]. In addition, the alternation of Hippo pathway components in epithelial cell lineage results in impaired lung development[15,16].

Here, we have investigated the function of pulmonary pericytes in the postnatal lung vasculature using inducible genetic experiments in mice in combination with three-dimensional imaging of thick sections at high resolution. We have made use of *Pdgfrb (BAC)-CreERT2* transgenic mice, which allow tamoxifen-inducible Cre-mediated recombination specifically in PDGFRβ-expressing cells. *Pdgfrb(BAC)-CreERT2*-mediated inactivation of the murine *Yap1* and *Wwtr1* genes led to impaired alveolar development by differentially altering the hepatocyte growth factor (HGF)/c-Met signalling pathway in epithelial cells, and angiopoietin-1/Tie2 signalling in ECs. Accordingly, the inactivation of the gene encoding angiopoietin-1 in PDGFRβ+ cells also impaired postnatal alveologenesis. Our findings demonstrate that pericytes have crucial, tissue-specific properties and help to orchestrate organ morphogenesis.

## Results

### Characterisation of pulmonary pericytes during lung development.
To study alveolar development and the role of pericytes in this process, we performed immunostaining and imaging of thick lung sections (200–300 μm). Combined RAGE and PECAM1 staining, which labels AT1 cells lining the inner surface of the lung and parenchymal ECs, respectively, showed that small epithelial saccules have formed at postnatal day (P) 4 (Supplementary Fig. 1a–c). At P21, the size of terminal airspaces is transiently decreased during alveolarization, which is followed by their expansion by P50 reflecting the growth of lung parenchyma[17] (Supplementary Fig. 1a, c). Coordinated development of a densely interconnected and flat capillary plexus can be also observed, which is highly distinct in its morphology from vessels in other organs[18] (Supplementary Fig. 1b, c; Supplementary Movie 1). Three-dimensional high-resolution images reveal the presence of pericytes, identified by expression of platelet-derived growth factor receptor β (PDGFRβ), in close association with capillaries but also with AT1 and AT2 epithelial cells in 4-week-old lungs (Fig. 1a–d and Supplementary Fig. 1d, Supplementary Movie 2). Quantitative analysis of ECs, visualized by expression of membrane-anchored tdTomato fluorescent protein and nuclear H2B-green fluorescent protein (GFP) fusion protein in *Cdh5-mT/nG* transgenic mice, and PDGFRβ+ pericytes at different developmental stages indicate a relatively stable ratio of ECs to pericytes of 7:1 to 9:1 (Fig. 1e and Supplementary Fig. 1e, f). Making use of PDGFRβ (encoded by the *Pdgfrb* gene) as a pericyte marker, we visualized the morphology and localization of these cells after irreversible genetic labelling with *Pdgfrb(BAC)-CreERT2* transgenic mice[19]. Following administration of tamoxifen from P1 to P3, *Pdgfrb(BAC)-CreERT2*-induced expression of GFP under control of the *R26-mT/mG* Cre reporter labels PDGFRβ+ pericytes in tight association with PECAM1+ ECs at P7, P21, and P50 (Fig. 1f–h and Supplementary Fig. 1g–i). Limiting the number of Cre recombination events by single administration of a low dose of 4-hydroxytamoxifen (4-OHT) enables the visualization of morphological changes in GFP+ pericytes, which extend numerous short cellular protrusions by P7 but display fewer, thinner, and longer processes at later stages (Supplementary Fig. 1h). Further characterization of *Pdgfrb(BAC)-CreERT2 R26-mT/mG* double transgenic mice confirmed that tamoxifen administration at P1–P3 did not lead to GFP expression in PDGFRα+ fibroblasts and alpha smooth muscle actin (αSMA)+ bronchial smooth muscle cells or myofibroblasts (Fig. 1i–k and Supplementary Fig. 2a–c). Quantitative reverse transcription PCR (RT-qPCR) analysis shows a strong enrichment of transcripts characteristic for pericytes, namely *Pdgfrb*, *Cspg4*, and *Notch3*, in GFP+ cells isolated from *Pdgfrb(BAC)-CreERT2 R26-mT/mG* lungs (Fig. 1l; Supplementary Fig. 2d and 3a, b). By contrast, expression of transcripts for *Pecam1* (encoding an endothelial junction protein) and *Sftpc* (the mRNA for surfactant protein C) is very low arguing against *Pdgfrb(BAC)-CreERT2*-mediated recombination in ECs or epithelial cells (Fig. 1l and Supplementary Fig. 3a). Coexpression of NG2 and

**Fig. 1** Characterisation of pulmonary pericytes during alveologenesis. **a**, **b** Three-dimensional reconstruction (**a**) and high magnification thin optical section (**b**) of confocal images with Airyscan detection showing AQP5-stained type 1 alveolar epithelial cells (green), PDGFRβ-stained pulmonary pericytes (PCs) (red) and PECAM1-stained endothelial cells (ECs) (blue) in lung at 4 weeks. Panels on the right show higher magnification of corresponding insets (**a**) or single channels (**b**). Scale bar, 30 μm (**a**, left), 15 μm (**a**, right), 10 μm (**b**, left) and 5 μm (**b**, right panels). **c**, **d** Three-dimensional reconstruction (**c**) and thin optical section (**d**) of confocal images showing SFTPC-stained, cuboidal AT2 cells (green), PDGFRβ-stained PCs (red) and RAGE-stained AT1 cells (blue) in lung at 4 weeks. Panels on the right show higher magnification of corresponding insets. Scale bar, 20 μm (left), 10 μm (right). **e** Quantitation of EC to PC ratio as measured in peripheral lung sections at different developmental stages. Data represents mean ± s.e.m. (*n* = 5 for P7, *n* = 6 for P21 and Adult mice). **f** Scheme showing the time points of tamoxifen administration (P1-3) and analysis for the *Pdgfrb(BAC)-CreERT2 R26-mT/mG* mice. **g–k** High magnification images of *Pdgfrb(BAC)-CreERT2 R26-mT/mG* lung sections at indicated stages showing pulmonary GFP+ PCs (green), PECAM1+ ECs, PDGFRβ+ PCs, PDGFRα+ fibroblasts or αSMA+ bronchial smooth muscle cells/myofibroblasts (red, as indicated). Arrows in **g** indicate PC cell bodies, in **g–k** GFP-positive PDGFRβ+ cells (**g**, **h**), GFP-negative PDGFRα+ cells (**i**) or αSMA+ cells (**j**, **k**) are marked. Scale bar, 20 μm (**g**, **h**), 15 μm (**i**, **k**) and 30 μm (**j**). **l** RT-qPCR analysis of *Pdgfrb* (pericytes), *Pecam1* (endothelium), and *Sftpc* (epithelium) expression in freshly sorted GFP+, CD31+ or EpCAM+ cells from P7 *Pdgfrb (BAC)-CreERT2 R26-mT/mG* lung. Data represents mean ± s.e.m. (*n* = 4 mice)

Notch3 in pulmonary pericytes is also confirmed by immunostaining in *Pdgfrb-CreERT2 R26-mT/mG* and *Cspg4-dsRed.T1* reporter mice (Supplementary Fig. 3c, d). Therefore pulmonary pericytes dynamically change their morphology in the course of lung development and these cells can be genetically modified and tracked with *Pdgfrb-CreERT2* transgenic mice.

**Defective alveologenesis in pericyte-specific *Yap1,Wwtr1* knockout mice**. The Hippo signalling pathway is a potent regulator of cell growth, differentiation, and tissue homeostasis, and has been implicated in the development of the pulmonary epithelium and tumourigenesis[13,14]. Two core components of the Hippo pathway, the transcriptional coregulators YAP1 and TAZ,

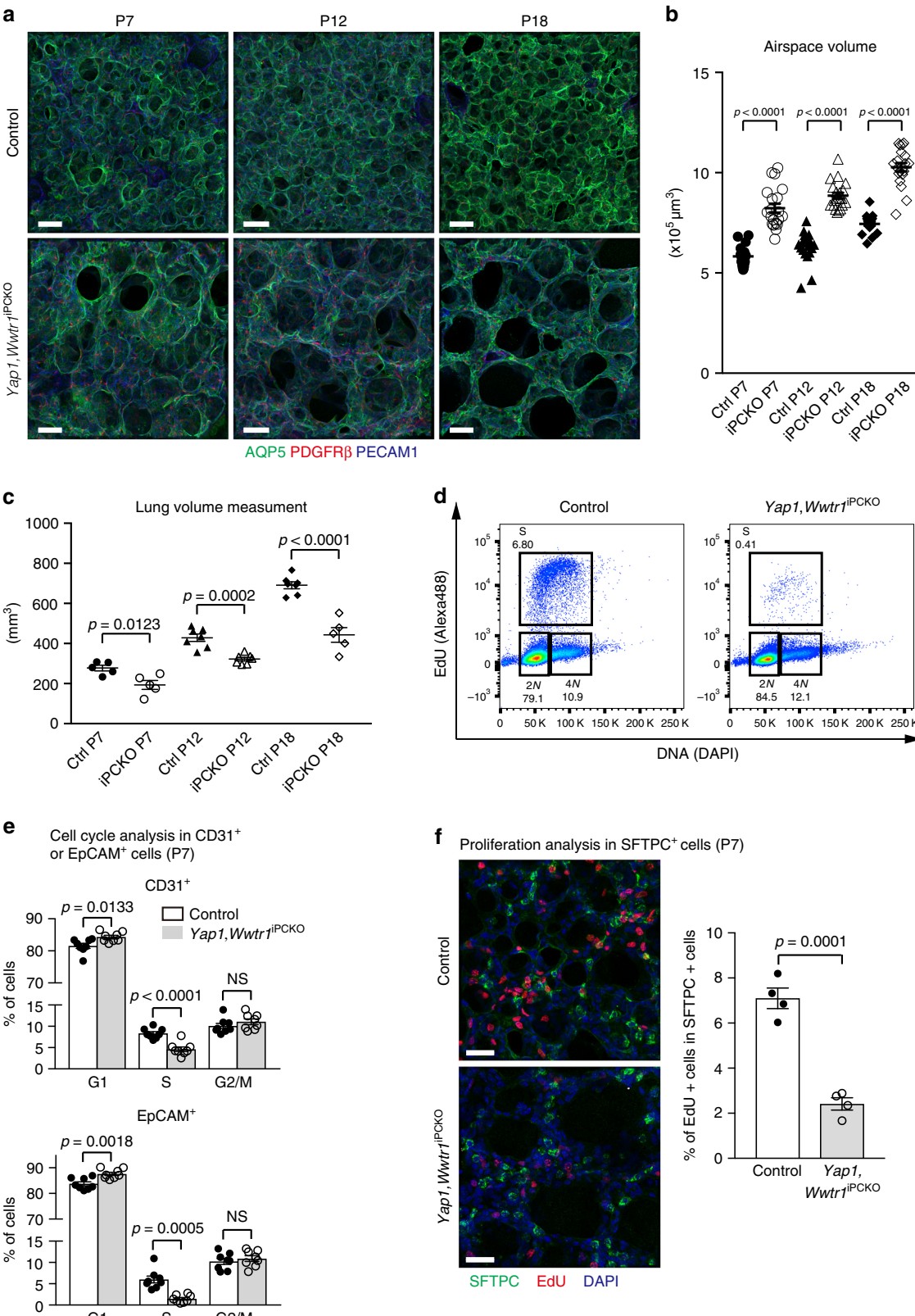

are expressed by pulmonary pericytes (Supplementary Fig. 4a, c). To analyse the function of YAP1 and TAZ in pulmonary pericytes, *Pdgfrb(BAC)-CreERT2* mice were interbred with mice carrying loxP-flanked alleles of *Yap1* or *Wwtr1*. The resulting *Yap1*[iPCKO] or *Wwtr1*[iPCKO] single mutants do not display a prominent lung phenotype, although YAP1 or TAZ immunostaining in PDGFRβ+ pericytes is strongly diminished (Supplementary Fig. 4b, d–f). To exclude the possibility of functional redundancy, we also generated *Yap1,Wwtr1*[iPCKO] double mutants. Western blot analysis revealed strong reductions in YAP1 and WWTR1 protein levels in P12 *Yap1,Wwtr1*[iPCKO] relative to littermate control total lung lysates despite the expression of both gene products by multiple cell types (Supplementary Fig. 4g). *Yap1, Wwtr1*[iPCKO] double mutant lungs display impaired alveolarization and a severe emphysema-like morphology (Fig. 2a and Supplementary Fig. 4h, i). A marked increase in airspace volume and a significant reduction of lung volume is already seen in mutants at P7, while mutant body weight is only slightly decreased. The weight difference between mutants and littermate controls increases gradually during postnatal growth leading to a substantial difference at P18 (Fig. 2b, c and Supplementary Fig. 5a). However, *Yap1,Wwtr1*[iPCKO] double mutants show no obvious change in vital functions such as arterial oxygen saturation and respiratory rates (Supplementary Fig. 5b). Reflecting the reduced *Yap1,Wwtr1*[iPCKO] lung volumes, we observed an accumulation of CD31+ ECs and EpCAM+ epithelial cells in G1 phase with a corresponding decrease in the percentage of cells in S phase, indicative of reduced cell proliferation (Fig. 2d, e and Supplementary Fig. 5c). Likewise, there is a notable decrease in AT2 cell proliferation, as measured by combined SFTPC and EdU staining (Fig. 2f). By contrast, activated caspase 3 staining, reflecting apoptosis, is not enhanced in *Yap1,Wwtr1*[iPCKO] lungs (Supplementary Fig. 5d). Recent studies have reported that AT2 cells continue to proliferate during postnatal alveologenesis and also include a population of stem cells that slowly generate new alveolar cells throughout adult life[20,21], while AT1 proliferation ends within the first week after birth. Immunostaining analysis with NKX2.1, a lung lineage transcription factor expressed by both AT1 and AT2 cells, and LAMP3 (an AT2 cell marker) shows that the proportion of AT2 cells is already slightly decreased in P7 *Yap1,Wwtr1*[iPCKO] lungs compared to controls, and this difference is notably exacerbated in P12 lungs (Fig. 3a, b). *Yap1,Wwtr1*[iPCKO] pericytes at P12, labelled by GFP expression from a *R26-mT/mG* reporter allele, exhibit narrow and thin processes similar to those seen in more mature wild-type pulmonary pericytes (Fig. 3c and Supplementary Fig. 1h). Flow cytometry analysis of *Yap1, Wwtr1*[iPCKO] PDGFRβ+ cells shows no significant change in abundance relative to control littermates (Fig. 3d and Supplementary Fig. 5e). Furthermore, immunostaining of PECAM1, ICAM2, and Erg in P12 *Yap1,Wwtr1*[iPCKO] lungs indicates decreased capillary formation in secondary septa and a reduction

of the pulmonary vasculature (Fig. 3e–h; Supplementary Movies 3, 4).

It is known that the differentiation of myofibroblasts and their production of elastin are important for the proper formation of secondary septa[22]. Immunostaining of the myofibroblast markers αSMA and tropoelastin, however, shows no significant changes between control and *Yap1,Wwtr1*[iPCKO] lungs (Fig. 3i, j), which is confirmed by immunoblotting of P7 total lung lysates (Fig. 3k). Consistent with the absence of *Pdgfrb(BAC)-CreERT2* activity in αSMA+ myofibroblasts in lung, these data argue that impaired alveologenesis in *Yap1,Wwtr1*[iPCKO] mutants is not caused by myofibroblast defects.

**Gene expression changes in YAP1/TAZ-deficient pulmonary pericytes.** To gain more insight into the molecular alterations in *Yap1,Wwtr1*[iPCKO] pericytes, we introduced an *Rpl22HA* allele, which enables the immunoprecipitation of actively translating polyribosome-bound mRNA from Cre-recombined cells[23]. RNA isolated and sequenced from control and mutant lungs in triplicate with this approach showed a strong enrichment of pericyte markers relative to input, whereas transcripts encoding endothelial and epithelial markers were depleted (Supplementary Fig. 6a). Differential gene expression analysis with an FDR-adjusted $p$-value $< 0.01$ and an absolute log2 fold change $> 0.5$ identifies 1309 differentially expressed genes, of which 829 are upregulated and 480 downregulated (Fig. 4a). DAVID Functional Annotation gene ontology analysis indicates that YAP1/TAZ-dependent gene changes are associated with cell migration, cell surface receptor signalling, and vascular development (Fig. 4b). To investigate how pericytes might control alveolarization, we focused on potential paracrine regulators. Loss of YAP1 and TAZ in PDGFRβ+ pericytes reduces the expression of several growth factors in these cells, including *Angpt1*, *Tgfb2*, *Wnt11*, *Bmp4*, and *Hgf* (Fig. 4c and Supplementary Fig. 6b), many of which have been previously implicated with high confidence in BPD[24]. An analysis of pericytes from different organs indicates that the growth factor expression profiles of these cells are tissue-specific and are therefore likely to reflect specialized functional roles (Fig. 4d).

**Alteration of HGF/c-Met signalling pathway in YAP1/TAZ-deficient pulmonary pericytes.** First, we investigated the HGF signalling pathway, as it is known that signalling by HGF and its receptor c-Met is important for lung morphogenesis[25,26]. At P7, *Hgf* expression predominates in pulmonary pericytes, while *Met* transcripts are expressed in epithelial cells (Fig. 4e). Characterization of c-Met immunostaining indicates that the receptor is mainly expressed by SFTPC+ cells in P12 lungs (Fig. 4f and Supplementary Fig. 6c). In *Yap1,Wwtr1*[iPCKO] mutants, expression of *Hgf* in pericytes is reduced and levels of phosphorylated c-Met, but not of total c-Met, are substantially lower in total lung

**Fig. 2** Altered alveologenesis in PC-specific *Yap1 and Wwtr1* mutant mice. **a** Three-dimensional reconstruction confocal images of *Yap1,Wwtr1*[iPCKO] and littermate control lungs stained for AQP5 (green), PDGFRβ (red), and PECAM1 (blue) at the indicated stages. Scale bar, 50 μm. **b** Quantitation of airspace volume in *Yap1,Wwtr1*[iPCKO] and control lung sections with three-dimensional reconstruction surface images. Data represents mean ± s.e.m. ($n = 4$ mice; unpaired two tailed student $t$-test or Welch's $t$-test) **c** Lung volume measurement of *Yap1,Wwtr1*[iPCKO] and littermate control lungs at the indicated stages. Data represents mean ± s.e.m. ($n = 5$ for P7, $n = 7$ for P12, $n = 7$ for P18 controls, $n = 5$ for P18 mutant mice, two-tailed unpaired $t$-test). **d** Representative flow cytometry plots of EdU incorporation in EpCAM + cells from P7 *Yap1,Wwtr1*[iPCKO] and littermate control mice. **e** Diagrams showing flow cytometry cell cycle analysis of CD31+ or EpCAM+ cells in P7 *Yap1,Wwtr1*[iPCKO] and littermate control lungs. Data represents mean ± s.e.m. ($n = 8$ mice, NS: not significant, Unpaired two tailed student $t$-test or Welch's $t$-test). **f** Three-dimensional reconstruction confocal images of P7 *Yap1,Wwtr1*[iPCKO] and littermate control lungs stained for SFTPC (green), EdU (red), and DAPI (blue). Quantitation of EdU positive and SFTPC positive cells is shown on the right. Data represents mean ± s.e.m. ($n = 4$; unpaired two tailed student $t$-test). Scale bar, 30 μm

lysates (Fig. 4g, h). Treatment of cultured pericytes with Verte-porfin, which inhibits YAP1/TAZ function by preventing nuclear translocation[27], suppresses the expression of *Hgf* in pericytes in vitro in a dose-dependent fashion without cell toxicity (Fig. 4i and Supplementary Fig. 6d, e). Likewise, HGF release by pericytes into the culture medium is also strongly reduced (Fig. 4i), which established that HGF production by pericytes is perfusion-independent.

**Pericyte-derived Angpt1 regulates alveologenesis.** In addition to HGF/c-Met signalling, angiopoietin-1 (Angpt1), one of the growth factors that is highly expressed by pulmonary pericytes (Fig. 4d), and the corresponding receptor Tie2/Tek, which is expressed by ECs (Fig. 5a), are involved in lung morphogenesis[28]. Expression of Angpt1 by PDGFRβ+ pulmonary pericytes is also confirmed with *Angpt1*[GFP] knock-in reporter mice (Fig. 5b). Whereas levels of *Angpt1* are dramatically reduced in *Yap1*,

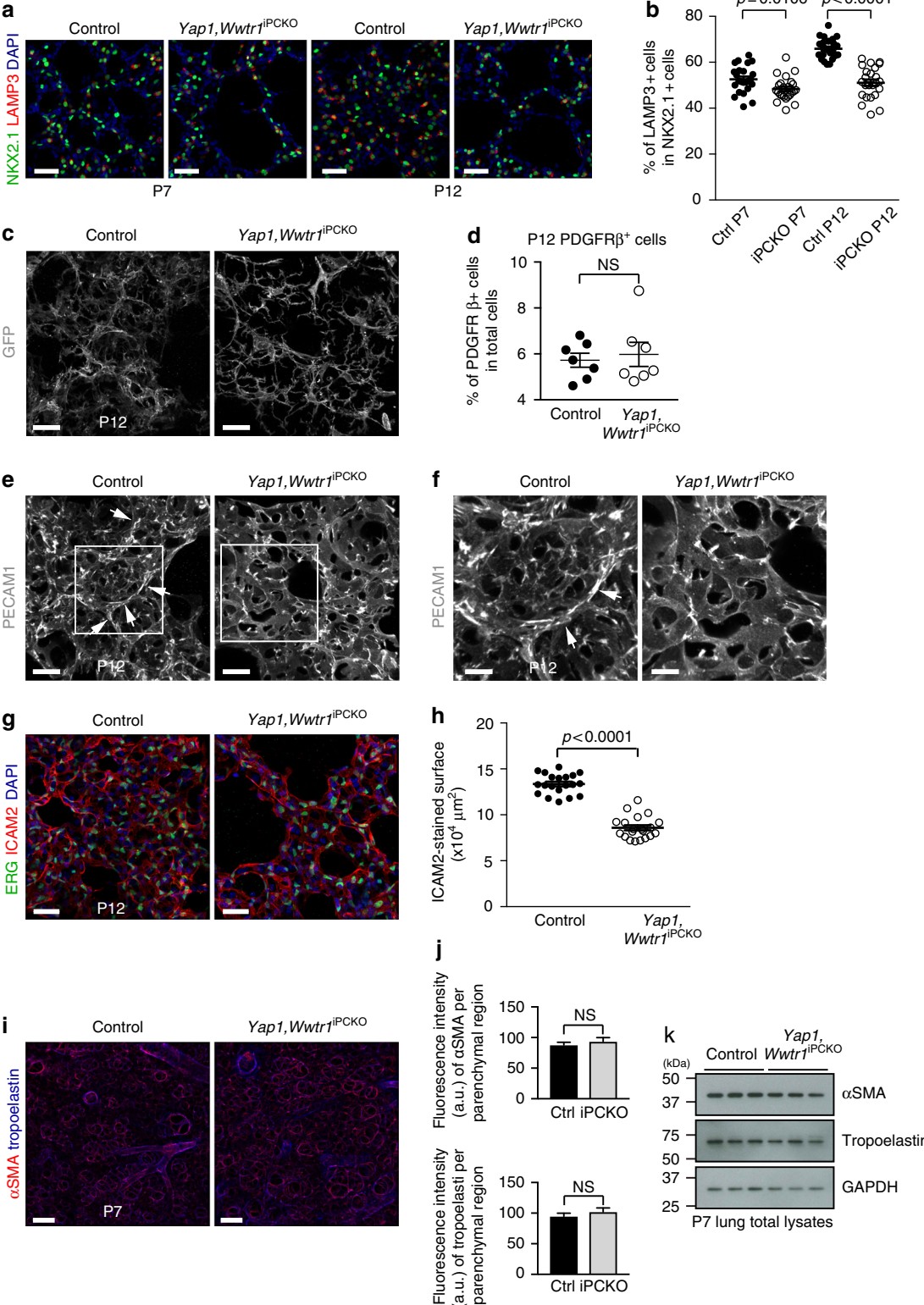

*Wwtr1*[iPCKO] PDGFRβ+ cells, as well as in cultured wild-type pericytes treated with Verteporfin or after siRNA-mediated depletion of Tead1/2/3/4, the expression of other components of this pathway is not altered in pulmonary ECs (Fig. 5c–e, and Supplementary Fig. 7a, b). A previously known TEAD binding motif can be identified in the upstream sequence of the *Angpt1* gene. To test the functionality of this DNA element, a fragment harbouring the −40 kb TEAD binding site was inserted into a luciferase reporter. This sequence leads to an increase in luciferase reporter expression, which is reduced upon mutation of the TEAD-binding DNA motif (Supplementary Fig. 7c). This result argues that a YAP/TAZ-TEAD complex transcriptionally regulates the expression of *Angpt1*.

Western blot analysis of *Yap1,Wwtr1*[iPCKO] lung lysates confirms strongly decreased expression of Angpt1 protein and significantly lower levels of phosphorylated Tie2 relative to control samples (Fig. 5f–h). As global *Angpt1* knockout mice are embryonic lethal[29], lungs from inducible *Pdgfrb(BAC)-CreERT2*-mediated *Angpt1* mutants (*Angpt1*[iPCKO]) were analysed (Fig. 5i). *Angpt1*[iPCKO] lungs copy major aspects of the *Yap1,Wwtr1*[iPCKO] phenotype such as alveologenesis defects, reduced capillary formation in the secondary septa and increased airspace volume (Fig. 5j–l, and Supplementary Fig. 7d). Like *Yap1,Wwtr1*[iPCKO] mutants, *Angpt1*[iPCKO] mice show no significant differences in myofibroblasts and tropoelastin deposition (Fig. 5m, n). Interestingly, expression of *Hgf* is also reduced in *Angpt1*[iPCKO] mutant pericytes (Supplementary Fig. 7e). It was previously shown that Angpt1 induces HGF expression in cultured endothelial cells and thereby promotes smooth muscle cell migration[30]. We found that treatment of cultured primary lung pericytes with high concentrations of COMP-Ang1, a soluble and stable variant of Angpt1, induces the expression of *Hgf* (Supplementary Fig. 7f). As transcripts encoding the angiopoietin-1 receptor Tie2 were strongly expressed in CD31+ ECs but not in pericytes isolated from P7 lung, we explored whether COMP-Ang1 might act through another receptor such as the integrin αvβ5, which was shown to mediate Angpt1-induced effects in astrocytes[31]. Consistent with a role of integrin αvβ5, treatment with anti-integrin αv blocking antibody completely suppresses the Angpt1-triggered increase in *Hgf* expression in cultured pulmonary pericytes (Supplementary Fig. 7g). Based on these findings, we also tested whether treatment with recombinant COMP-Ang1 might restore alveologenesis in *Yap1,Wwtr1*[iPCKO] mice. Remarkably, repeated COMP-Ang1 administration at P4, P6, P8, and P10 leads to partially but significantly improved alveologenesis in P12 *Yap1,Wwtr1*[iPCKO] mice (Supplementary Fig. 7h, i). The sum of these findings indicates that Angpt1 regulates *Hgf* expression in pericytes cell-autonomously

and thereby plays an important role in postnatal lung morphogenesis.

**Substrate stiffness controls YAP1/TAZ localization.** Next, we investigated the upstream factors controlling YAP1/TAZ activity in pulmonary pericytes. Mst1/Stk4 and Mst2/Stk3 are part of the canonical Hippo pathway and negative regulators of YAP1/TAZ activity[13]. However, postnatal *Pdgfrb(BAC)-CreERT2*-mediated ablation of *Stk3* and *Stk4* genes does not affect normal lung morphogenesis (Supplementary Fig. 8a, b). Mechanical properties of tissues, such as its elasticity or stiffness, are pivotal regulators of morphogenesis[32] and mechanotransduction has been implicated in the regulation of YAP1/TAZ activity[33]. Interestingly, several genes related to tissue microelasticity are upregulated at P21 in pulmonary pericytes (Supplementary Fig. 8c), suggesting that local tissue stiffness increases during alveolar morphogenesis. Consistent with the regulation of lung pericytes by biomechanical parameters, YAP1/TAZ are mostly cytoplasmic in these cells when cultured on soft substrates, whereas growth on stiff substrates leads to nuclear accumulation of YAP1/TAZ and the extension of pericyte protrusions similar to those seen in vivo (Supplementary Fig. 1h and 8d, e).

Together, our results establish that pericytes act as important morphogenetic regulators with specialized, presumably organ-specific properties. In postnatal lung, the paracrine signalling capability of pericytes, which is controlled by YAP1/TAZ and mechanical factors, is crucial for the interplay with AT2 epithelial cells and thereby alveologenesis (Fig. 5o).

## Discussion

The lung is thought to comprise as many as 40 different cell types including several epithelial cell subpopulations. Squamous AT1 alveolar cells cover most of the internal surface of the pulmonary tree, whereas cuboidal AT2 cells are confined to the corners of the alveolus. In addition to their important role in the production, secretion, and recycling of pulmonary surfactant[34], postnatal AT2 cells have stem cell-like roles, can give rise to AT1 cells and contribute to alveolar renewal, lung repair, and cancer[20,21]. During development, both AT1 and AT2 cells arise from a bipotent progenitor[20,35]. Other studies have identified bronchioalveolar stem cells (BASCs) as the source of bronchiolar and alveolar cells during injury repair and as a potential target in disease processes[9]. Multiple pathways, including epithelial growth factor (EGF), bone morphogenetic protein (BMP), fibroblast growth factor (FGF), and HGF signalling, have been proposed to control the properties of pulmonary epithelial stem and progenitor cells. High expression of c-Met has been reported both in

---

**Fig. 3** Lung phenotype of PC-specific *Yap1,Wwtr1* mutants. **a** Maximum intensity projections of P7 and P12 *Yap1,Wwtr1*[iPCKO] and littermate control lungs showing NKX2.1-stained (green) AT1 and AT2 cell nuclei, LAMP3-stained AT2 cells (red), and DAPI (blue). Scale bar, 30 μm. **b** Quantitation of NKX2.1 positive and LAMP3 positive cells shown in **a**. Data represents mean ± s.e.m. (*n* = 4 mice; two-tailed unpaired *t*-test or Welch's *t*-test). **c** Three dimensional high magnification reconstruction images of *Yap1,Wwtr1*[iPCKO] *R26-mT/mG* and littermate control lungs stained for GFP (white). Scale bar, 20 μm. **d** Flow cytometric analysis of PDGFRβ+ pericytes in P12 *Yap1,Wwtr1*[iPCKO] and control lungs. Data represents mean ± s.e.m. (*n* = 7 mice; NS not significant, two-tailed unpaired *t*-test). **e, f** Confocal images of P12 *Yap1,Wwtr1*[iPCKO] and littermate control lungs stained for PECAM1 (white). Note formation of capillaries at the secondary septa in controls (arrowheads) but not in mutants. Panels in **f** show higher magnification of insets in **e**. Scale bar, 20 μm (**e**) and 10 μm (**f**). **g** Three-dimensional high magnification reconstruction images of P12 *Yap1,Wwtr1*[iPCKO] and littermate control lungs stained for ERG (green), ICAM2 (red), and DAPI (blue). Scale bar, 30 μm. **h** Quantitation of ICAM2-stained vascular density based on 3D reconstruction surface images, as shown in **g**. Data represents mean ± s.e.m. (*n* = 4 mice; unpaired two tailed student *t*-test) **i** Maximum intensity projections of P7 *Yap1,Wwtr1*[iPCKO] and littermate control lungs stained for αSMA (red) and tropoelastin (blue). Scale bar, 100 μm. **j** Quantitation of staining intensity for αSMA or tropoelastin shown in **i**. Signal intensity was normalized to the size of the parenchymal region. Data represents mean ± s.e.m. (*n* = 4 mice; NS not significant, two-tailed unpaired *t*-test). **k** Western blot analysis of αSMA and tropoelastin in P7 *Yap1,Wwtr1*[iPCKO] and littermate control total lung lysates (*n* = 3 mice). Molecular weight marker (kDa) is indicated

BASC and primary AT2 cells from rat lung[36], which is consistent with our own data showing prominent c-Met immunostaining of SFTPC+ cells in P12 lung. HGF/c-Met signalling mediates the proliferation and branching morphogenesis of lung epithelial cells in vitro[36,37] and, as the administration of neutralizing anti-HGF antibody or truncated (soluble) c-Met protein show, is also essential for alveologenesis in neonatal rats[38]. Mice with an AT2 cell-specific, postnatal deletion of *Met* show impaired alveolar formation and reduced AT2 cells[26], which is consistent with our

findings in *Yap1,Wwtr1*[iPCKO] lungs. HGF expression in lung has been previously attributed to lung mesenchymal cells and fibroblasts[39,40]. Our own data now argue that pericytes are a critical source of HGF in the postnatal lung and, accordingly, *Pdgfrb(BAC)-CreERT2*-mediated loss of Yap1 and TAZ dramatically reduces HGF expression in PDGFRβ+ cells and thereby levels of active (phosphorylated) c-Met in total lung lysates. These data indicate that postnatal pulmonary pericytes are a key source of growth factor signals controlling the behaviour of epithelial

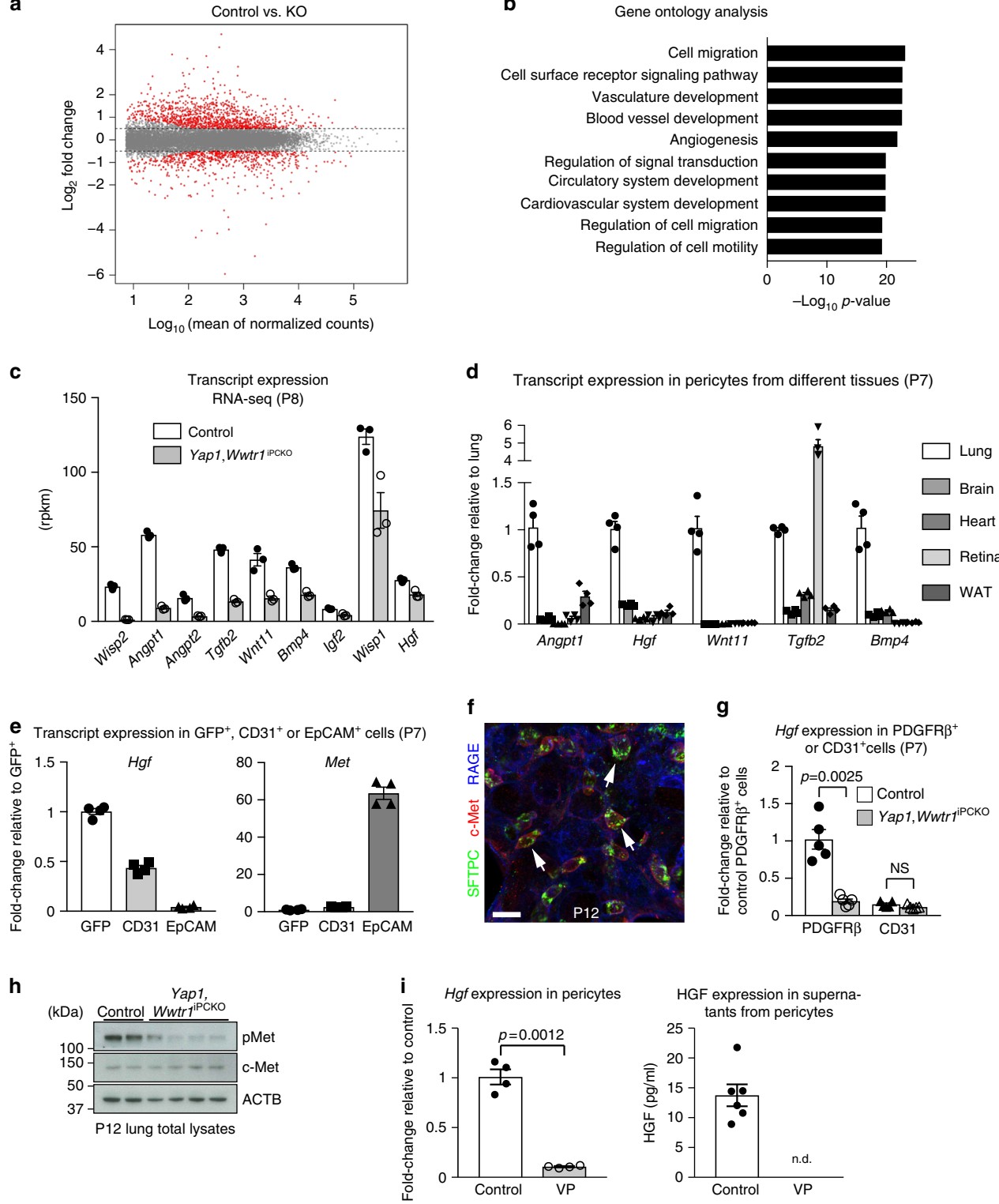

progenitor cells, which might be also relevant in the context of disease processes. It has been reported that low HGF concentrations in tracheal aspirate fluid samples of human neonates are associated with more severe BPD, a common chronic lung disease affecting premature infants[41]. Exome-sequencing has also identified nonsynonymous mutations in the human *HGF* gene as a genetic factor predisposing to BPD[24]. In addition, treatment with recombinant HGF improves lung defects in the hyperoxia-induced neonatal mice model of BPD[42]. Mice lacking c-Met in AT2 epithelial cells show an increase in oxidative stress, which is relevant to chronic obstructive pulmonary disease (COPD)[26,43]. Reduced HGF secretion in peripheral lung of COPD patients correlates with disease severity[44]. Future work should therefore consider pulmonary pericytes in the context of pathologically altered HGF production.

Signalling by angiopoietins and the receptor tyrosine kinase Tie2 plays important roles in the regulation of angiogenesis, EC survival, and vascular integrity[45,46]. While expression of angiopoietin-1 is frequently attributed to pericytes and mural cells[47,48], we found that this is indeed the case for lung but, unexpectedly, not for brain and retina at P7. Moreover, pericyte-derived angiopoietin-1 is indispensable for alveologenesis and is also likely to control EC behaviour in postnatal lung, which would help to coordinate multiple different cell types during pulmonary development. Paracrine (also termed angiocrine) signalling has been previously associated with ECs in the control of organ morphogenesis and regeneration[2-4,49]. Unexpectedly, pericytes emerge from our study as novel regulators of tissue growth and patterning in addition to their acknowledged roles in the maintenance of vessel integrity. We also found that pericytes express distinct growth factor signatures in an organ-specific fashion, which may well reflect distinct functional roles in the regulation of morphogenetic processes in the surrounding tissue. Thus, pericytes are not only safeguarding vascular integrity but are also specialized mediators of tissue morphogenesis with potential relevance for pathobiological processes and regeneration.

## Methods

**Mouse models.** C57BL/6J mice were used for the analysis of wild-type lung. Lineage tracing experiments were performed by mating *Pdgfrb(BAC)-CreERT2*[19,50] and *R26-mT/mG* reporter mice[51]. Cre activity and GFP expression were induced by intraperitoneal injection of 50 μg tamoxifen (T5648, Sigma) in ethanol/Kolliphor EL (C5135, Sigma) from postnatal day (P) 1 to 3 or 1 μg 4-hydroxy tamoxifen (4-OHT) in ethanol/Kolliphor EL (H7904, Sigma) at P2 for a low dosage. For inducible genetic experiments, *Pdgfrb(BAC)-CreERT2* transgenics were interbred with mice carrying loxP-flanked *Yap1* (*Yap1*[lox/lox]), *Wwtr1* (*Wwtr1*[lox/lox])[52], *Angpt1* (*Angpt1*[lox/lox])[53], *Stk3* (*Stk3*[lox/lox]), or *Stk4* (*Stk4*[lox/lox]) alleles[54]. Cre activity and gene inactivation were induced by intraperitoneal injection of 50 μg tamoxifen from P1 to 3. Tamoxifen was administered to both controls and mutants in all

inducible genetic experiments. At the age of P7, P12, and P18, lungs were collected from *Yap1*[lox/lox] *Wwtr1*[lox/lox] *Pdgfrb(BAC)-CreERT2*[T/+] (*Yap1,Wwtr1*[iPCKO]) mutants and Cre-negative *Yap1*[lox/lox] *Wwtr1*[lox/lox] littermate controls. P12 lungs were collected from *Yap1*[lox/lox] *Pdgfrb(BAC)-CreERT2*[T/+] (*Yap1*[iPCKO]), *Wwtr1*[lox/lox] *Pdgfrb(BAC)-CreERT2*[T/+] (*Wwtr1*[iPCKO]), *Angpt1*[lox/lox] *Pdgfrb(BAC)-CreERT2*[T/+] (*Angpt1*[iPCKO]) or *Stk3*[lox/lox] *Stk4*[lox/lox] *Pdgfrb(BAC)-CreERT2*[T/+] (*Stk3, Stk4*[iPCKO]) mutants and the corresponding Cre-negative littermate controls. *Yap1, Wwtr1*[iPCKO] mice were also interbred with *R26-mT/mG* reporters and the resulting *Yap1*[lox/lox] *Wwtr1*[lox/lox] *Pdgfrb(BAC)-CreERT2*[T/+] mutants and *Yap1*[lox/+] *Wwtr1*[lox/+] *Pdgfrb(BAC)-CreERT2*[T/+] controls were analysed at P12 after tamoxifen administration from P1 to P3. For RiboTag analysis, mice were interbred with *Rpl22*[tm1.1Psam] (*Rpl22*[lox/lox]) mice[23]. *Yap1*[lox/lox] *Wwtr1*[lox/lox] *Rpl22*[lox/+] *Pdgfrb (BAC)-CreERT2*[T/+] mutants and *Yap1*[lox/+] *Wwtr1*[lox/+] *Rpl22*[lox/+] *Pdgfrb(BAC)-CreERT2*[T/+] controls were analysed at P8 after tamoxifen administration from P0 to P2. For RT-qPCR or RNA-seq analysis of RiboTag samples, *Rpl22*[lox/lox] *Pdgfrb (BAC)-CreERT2*[T/+] mice were analysed at P2, P7 or P21 after 50 μg 4-OHT administration at P1. Tg(*Cspg4-DsRed.T1*)[1Akik][55] and *Angpt1*[tm1.1Sjm] reporter mice[56] were previously published. *Cdh5-mT/nG* transgenic mice were generated as follows: A cDNA encoding membrane-tagged tdTomato (Addgene plasmid # 17787), 2A peptide, AU1 tag and H2B-EGFP (Addgene plasmid # 11680) followed by a polyadenylation signal sequence and a FRT-flanked ampicillin resistance cassette were introduced by recombineering into the start codon of Cdh5 in PAC clone 353-G15. After Flp-mediated excision of the ampicillin resistance cassette in bacteria, the resulting constructs were validated by PCR analysis and used in circular form for pronuclear injection into fertilized mouse oocytes. Founders, identified by PCR genotyping, were screened by immunostaining with ERG antibody. Genotypes of mice were determined by PCR. Arterial oxygen saturation and respiratory rate were measured with a Collar Sensor by MouseOx Plus (Starr Life Sciences Corp, USA) as outlined in the manufacturer's instructions. For the administration of COMP-Ang1[57], *Yap1,Wwtr1*[iPCKO] and control littermates were injected with tamoxifen from P1 to P3 and with COMP-Ang1 (12.5 μg) or similar volume of PBS at P4, P6, P8, and P10, analysed at P12. All animal experiments were performed in compliance with the relevant laws and institutional guidelines, were approved by local animal ethics committees and were conducted with permissions granted by the Landesamt für Natur, Umwelt und Verbraucherschutz (LANUV) of North Rhine-Westphalia to the Max Planck Institute for Molecular Biomedicine or by the Animal Care and Use Committee of KAIST (No. KA2015-15). Animals were combined in groups for experiments irrespective of their sex.

**Lung sample preparation and immunohistochemistry.** Mice up to 12 days old were anesthetized with ketamine/xylazine, before the chest cavity was opened by cutting along the sternum and the ribs adjacent to the diaphragm to expose the heart and lungs. A warm (37 °C) solution of 6% gelatin (G1890, Sigma) in PBS was gently perfused through the right ventricle with manual pressure. Before taking out the needle from right ventricle, a small, ice-cold cotton pad was placed on the injected site to solidify the gelatin inside the heart. The ventral trachea was cannulated with an intravenous catheter tube. The lungs were inflated to full capacity by gently injecting warm (37 °C) 1% low gelling agarose (A4018, Sigma) in PBS. After agarose-inflation, the lungs were placed to ice-cold PBS. The lungs and heart were removed and placed in 2% paraformaldehyde solution (PFA in PBS, 4 °C, P6148, Sigma) for 30 min. Lungs from older mice (13 days or more) were processed as described except that following anesthesia, trachea was cannulated with an intravenous catheter tube that was secured by tying a suture around the trachea. The lungs were inflated with air followed by the perfusion of gelatin solution (~25 mmHg). The inflated lungs were deflated by releasing positive air pressure followed by re-inflation to total lung capacity with agarose solution (~20–25 cmH2O). After the exposure to ice-cold PBS to solidify the agarose, the cannula was withdrawn and the suture tightened.

**Fig. 4** Gene expression changes in *Yap1,Wwtr1*[iPCKO] pericytes. **a** MA-plots of differentially regulated genes between P8 *Yap1,Wwtr1*[iPCKO] and control pericytes. The x-axis represents the mean normalized counts and the y-axis shows the log2 fold change. Differentially regulated genes are represented by red colored points (FDR-adjusted P-value < 0.01 and absolute log2 fold change > 0.5). **b** Gene ontology analysis for significantly differentially regulated genes (RNA-seq analysis) in *Yap1,Wwtr1*[iPCKO] and control pericytes. **c** RiboTag and RNA-seq-based expression levels of the indicated transcripts relative to P8 control. RPKM (reads per kilobase of exon per million mapped reads) values obtained from RNA-seq are shown. Data represents mean ± s.e.m. (n = 3). **d** RT-qPCR analysis of indicated transcripts in immunoprecipitated (IP) RNA from P7 *Pdgfrb(BAC)-CreERT2 Rpl22HA* lung, brain, heart, retina, and inguinal white adipose tissues (WAT). Data represents mean ± s.e.m. (n = 4 mice). **e** RT-qPCR analysis of *Hgf* and *Met* expression in freshly sorted GFP+, CD31+ or EpCAM+ cells from P7 *Pdgfrb-CreERT2 R26-mT/mG* lungs. Data represents mean ± s.e.m. (n = 4 mice). **f** High magnification images of P12 lungs stained for SFTPC (green), c-Met (red), and RAGE (blue). Arrows indicate SFTPC+ c-Met+ cells. Scale bar, 15 μm. **g** *Hgf* expression in freshly sorted PDGFRβ+ or CD31+ cells from P7 *Yap1,Wwtr1*[iPCKO] and littermate control lungs. Data represents mean ± s.e.m. (n = 5 mice; NS not significant, Welch's t-test or two-tailed unpaired t-test). **h** Western blot analysis of total and phosphorylated c-Met (pMet) in P12 *Yap1,Wwtr1*[iPCKO] and control total lung lysates (n = 2 for controls and 4 for mutant mice). Molecular weight marker (kDa) is indicated. **i** RT-qPCR analysis of *Hgf* mRNA expression (left) and HGF levels in supernatants (right) from 1 μM Verteporfin (VP) treated cultured pericytes and controls at 48 h. Data represents mean ± s.e.m. (n = 4 and 6, Welch's t-test). n.d. not detectable

The lung lobes were separated and trimmed to create a flat surface, glued to a mounting block with cyanoacrylate glue (UHU, 48700), submerged in ice-cold PBS and sliced (200–300 μm) using vibrating blade microtome (VT1200, Leica). Lung slices were fixed in 4%PFA/PBS at 4 °C for 1 h, washed thoroughly in PBS and incubated twice in PBS at 55 °C for 30 min. After washing in PBS, blocking solution (5% donkey serum, 0.5% Triton X-100 in PBS) was applied for 30 min at room temperature and sections were treated with primary antibodies in blocking solution over night at 4 °C. Following two washes with PBS, sections were incubated with

secondary antibodies in blocking solution over night at 4 °C. After three wash steps with PBS, sections were mounted using FluoroMount-G (Southern Biotech) under cover slips sealed with nail polish that were spaced from the slide with tapes.

The following primary antibodies were used: rabbit anti-Aquaporin 5 (178615, Millipore, 1:200), rat anti-CD140b (14–1402, eBioscience, 1:100), goat anti-CD31 (FAB3628, R&D Systems, 1:200), rabbit anti-Prosurfactant Protein C (AB3786, Millipore, 1:200), chicken anti-GFP (GFP-1010, aves, 1:1000), rabbit anti-GFP conjugated to Alexa Fluor (AF) 488 (A21311, Thermo Fisher, 1:200), goat anti-

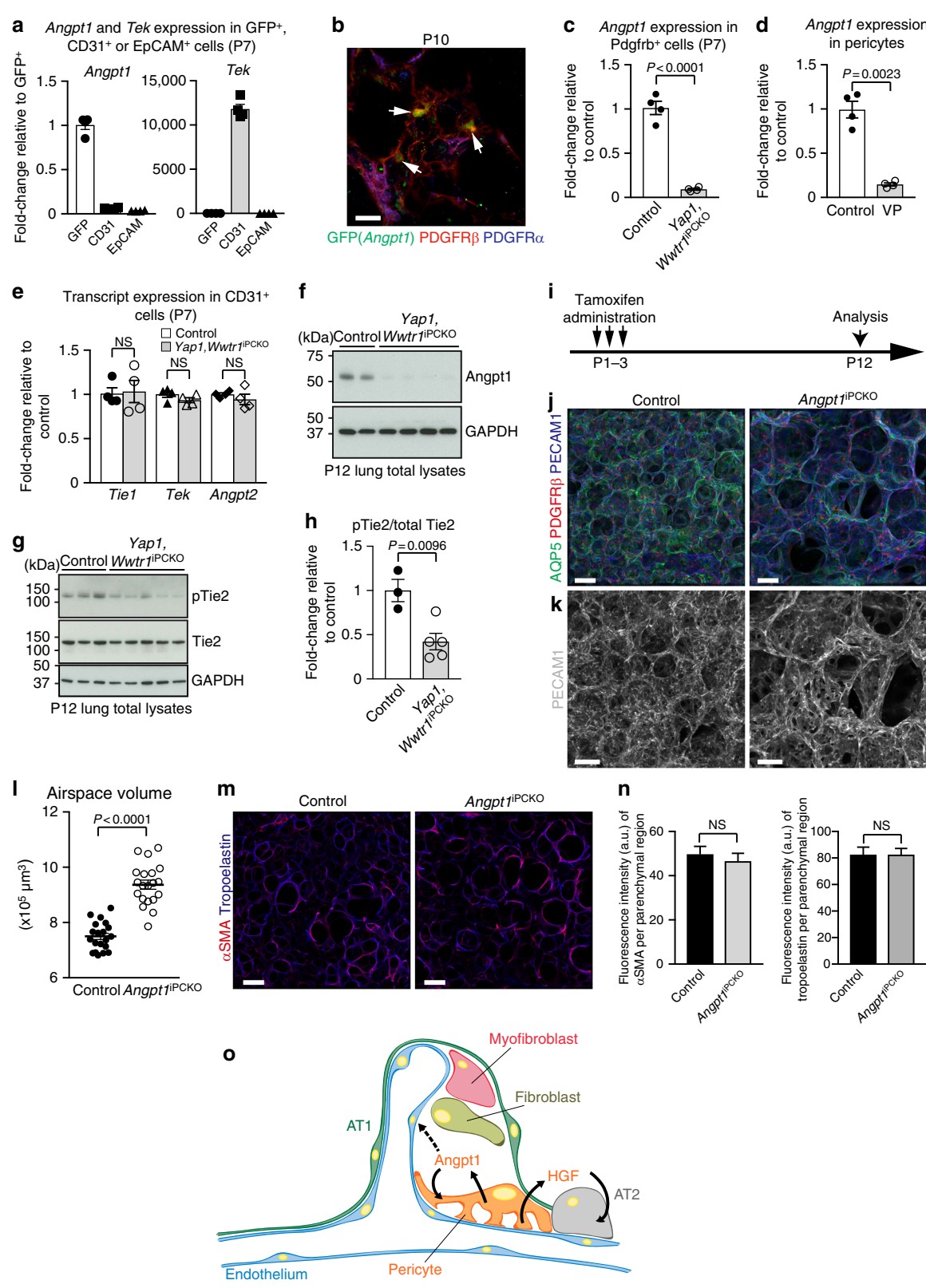

PDGFR alpha (AF1062, R&D Systems, 1:200), anti-α-SMA-Cy3 (C6198, Sigma, 1:200), anti-α-SMA eFluor 660 (C6198, Sigma, 1:200), rabbit anti-TTF1 (ab76013, abcam, 1:200), rat anti-DC-LAMP/CD208 (DDX0192, Dendritics, 1:200), rat anti-CD102 (553326, BD Biosciences, 1:100), rabbit anti-Tropoelastin (ab21600, abcam, 1:200), goat anti-HGF R/c-MET (AF527, R&D Systems, 1:200), rat anti-RAGE (MAB1179, R&D Systems, 1:200), rabbit anti-ERG (ab110639, abcam, 1:200), goat anti-PDGFR beta (AF1042, R&D Systems, 1:200), rabbit anti-NG2 (AB5320, Millipore, 1:100), goat anti-Notch3 (AF1308, R&D Systems, 1:200), rabbit anti-YAP (#14074, Cell Signaling, 1:100), rabbit anti-WWTR1 (HPA007415, Sigma, 1:200) and rabbit anti-Cleaved Caspase-3 (#9664, Cell Signaling, 1:100).

The following secondary antibodies were used: donkey anti-rabbit IgG conjugated to AF488 (A21206, Thermo Fisher, 1:400), donkey anti-chicken IgY AF488 (703-545-155, Jackson ImmunoResearch, 1:400), donkey anti-rat IgG AF488 (A21208, Thermo Fisher, 1:400), donkey anti-rat IgG Cy3 (712-165-153, Jackson ImmunoResearch, 1:400), donkey anti-rabbit IgG AF546 (A10040, Thermo Fisher, 1:400), donkey anti-rat IgG AF594 (A21209, Thermo Fisher, 1:200), donkey anti-rabbit IgG AF594 (A21207, Thermo Fisher, 1:200), donkey anti-goat IgG AF647 (A21447, Thermo Fisher, 1:200) and donkey anti-rabbit IgG AF647 (A31573, Thermo Fisher, 1:200). Nuclei were counterstained with DAPI (D9542, Sigma, 2 μg/ml). Proliferating cells were labelled in vivo by intraperitoneal injection of 33.3 μg of EdU (Thermo Fisher) per gram of body weight 90 min before sample collection. After thorough staining and washing, EdU-positive cells were stained with Click-iT EdU Alexa Fluor 647 Imaging Kit (C10340, Thermo Fisher) following manufacturer's instructions. Dissection, processing, sectioning, staining, imaging, and analysis for phenotypes of mutant mice and controls were always performed together under identical conditions.

The total lung volume was measured by the fluid displacement method[58].

**Image acquisition and quantitative analysis**. Images of lung sections were acquired using a Leica TCS SP8 confocal laser scanning microscope with LAS X software, a Zeiss LSM780/LSM880 confocal laser scanning microscope with online fingerprinting mode and ZEN software, or a Zeiss LSM880 with Airyscan detector. Z-stacks of images were processed and 3D reconstructed with Imaris software (version 7.7.2, Bitplane). Quantitation was performed with Imaris software or Volocity (Version 6.3, Perkin Elmer). Photoshop and Illustrator (version CS6, Adobe) were used for image processing. To determine the EC to pericyte ratio, the number of ECs and pericytes was calculated with Volocity software out of 45 to 54 fields sized 213 × 213 × 21 μm in five to six Cdh5-mT/nG transgenic lung samples per group. ECs were identified based on nuclear H2B-GFP and tdTomato signals, and pericytes were identified as PDGFRβ+ αSMA− and GFP−. The number of EdU and SFTPC positive cells was measured with Volocity software out of 24 fields sized 247 × 247 × 10 μm in four lung samples per group. The number of NKX2.1 and LAMP3 positive cells was measured with Imaris software out of 24 fields sized 213 × 213 × 20 μm in four lung samples per group. Vessel density was measured with Imaris out of 20 fields sized 213 × 213 × 21 μm in four lung samples per group based on 3D surface rendering images from 3D reconstructed images. Intensity of αSMA and tropoelastin staining that was normalized based on parenchymal region was measured with the Volocity out of 20 fields sized 213 × 213 × 17.7 μm in four lung samples per group and normalized to the lung parenchymal region. Airspace volume was measured with Imaris out of 20 or 24 fields sized 213 × 213 × 31 μm in four lung samples per group based on 3D surface rendering images from 3D reconstructed images. The number of active caspase 3 positive cells and total cells was measured with Imaris software out of 24 fields sized 213 × 213 × 30 μm in four lung samples per group. Fields containing major airway or vessel structures were not included in quantitation. All images shown are representative for the respective staining in several experiments. Within each experiment, laser excitation and confocal scanner detection were kept the same.

**Flow cytometry**. Mice were anesthetized, the chest cavity was opened, and ice-cold PBS was perfused through the right ventricle and the ventral trachea was cannulated. The lungs were inflated by injecting dispase solution (25 U/ml in PBS, 17105041, Thermo Fisher), removed, and other tissues and bronchi were trimmed. Lungs were minced in 3–5 ml of digestion buffer consisting of Liberase TM (100 μg/ml, 05401119001, Roche) and DNase I (0.1 mg/ml, DN25, Sigma) in MEM (31095-029, Gibco) supplemented with 25 mM HEPES and incubated with trituration for 30 min at 37 °C. An equal volume of PBS supplemented with 2% FCS was added and the cell suspension was passed through a 100 μm mesh filter followed by centrifugation at 330 g for 5 min. The precipitated cells were resuspended in red blood cell lysing buffer (R7757, Sigma), incubated for 4 min, passed through a 40 μm mesh filter, centrifuged at 330 g for 5 min, and finally resuspended in sorting buffer (PBS supplemented with 2% FCS). For cell sorting from Pdgfrb (BAC)-CreERT2 R26-mT/mG mice, the cells were immunostained with rat anti-TER-119-Pacific Blue (116232, BioLegend, 1:100), rat anti-CD45-Pacific Blue (103126, BioLegend, 1:100), rat anti-CD31-PerCP/Cy5.5 (102420, BioLegend, 1:50), rat anti-CD326-phycoerythrin (PE)/Cy7 (118216, BioLegend, 1:50) and rat anti-CD140a-allophycocyanin (APC) (17-1401-81, eBioscience, 1:100) for 30 min on ice. For cell sorting from Yap1,Wwtr1iPCKO mice and Angpt1iPCKO mice, cells were immunostained with rat anti-TER-119-Pacific Blue (116232, BioLegend, 1:100), rat anti-CD45-Pacific Blue (103126, BioLegend, 1:100), rat anti-CD31-FITC (RM5201, Thermo Fisher, 1:50), rat anti-CD140a-PE (12–1401–81, eBioscience, 1:100), rat anti-CD326-PE/Cy7 (1:50) and rat anti-CD140b-APC (17–1402–82, eBioscience, 1:25) for 30 min on ice. After washing, cells were treated with DAPI to exclude dead cells. Cell sorting was performed with a FACS Aria IIu (BD Biosciences) using a 70 μm nozzle and FACSDiva software.

Proliferating cells were labelled in vivo by intraperitoneal injection of EdU (Thermo Fisher) per gram of body weight 90 min before sample collection. Cell suspensions were prepared as described above and were immunostained with rat anti-CD31-Biotin (553371, BD Biosciences, 1:100), rat anti-CD140a-PE (1:100), rat anti-CD326-PE/Cy7 (1:50) and rat anti-CD140b-APC (1:25) for 30 min on ice. After washing, cells were stained with Streptavidin-BV605 (563260, BD Biosciences, 1:200) for 30 min on ice. After thorough washing, cells were stained with Click-iT Plus EdU Alexa Fluor 488 Flow Cytometry Kit (C10632, Thermo Fisher) following manufacturer's instructions. Finally, cells were again washed, treated with DAPI and acquired on FACS Aria IIu, followed by the analysis with the FlowJo software (FlowJo, LLC). Boundaries between positive and negative populations have been defined by using single stained and FMO (fluorescence minus one) controls.

**Cell culture and Luciferase assay and ELISA assays**. For pulmonary pericyte culture, freshly sorted CD140β+, CD140α−, CD326−, CD31−, TER-119−, and CD45− populations were cultured on 100 μg/ml collagen I (BD Biosciences) coated dishes in Pericyte Medium (#1201, ScienCell). Cultured cells were maintained at 37 °C in a humidified 5% CO2 atmosphere. More than 80% of cells are PDGFRβ+ and NG2+. Pericytes were transfected with siRNA by using Lipofectamine RNAiMAX Transfection Reagent (Thermo Fisher) according to the manufacturer's instructions. The following Silencer Select siRNAs were purchased from Thermo Fisher: mouse Tead1 s74929, mouse Tead2 s74931, mouse Tead3 s74934, and mouse Tead4 s74938. Silencer Select Negative Control No.1 siRNA (Thermo Fisher) was used for control transfection. Cells were harvested 72 h after transfection. For COMP-Ang1 treatment, cells were serum starved overnight before stimulation with bovine serum albumin (BSA; 1000 ng/ml) or 100, 500 or 1000 ng/ ml COMP-Ang1 for 24 h in the presence or absence of rat IgG or anti-Integrin Alpha V antibody (LS-C57970, LifeSpan BioSciences, 30 μg/ml), and harvested for RNA isolation. For Luciferase assay, we have searched for the known consensus motif for TEAD mouse genomic sequence up to 50 kbp from

**Fig. 5** Pericyte-derived Angpt1 controls alveologenesis. **a** RT-qPCR analysis of Angpt1 and Tie2/Tek expression in freshly sorted lung GFP+, CD31+ or EpCAM+ cells from P7 Pdgfrb(BAC)-CreERT2 R26-mT/mG mice. Data represents mean ± s.e.m. (n = 4 mice). **b** High magnification images of P10 Angpt1GFP lungs stained for GFP (green), PDGFRβ (red), and PDGFRα (blue). Arrows indicate GFP and PDGFRβ double positive pericytes. Scale bar, 15 μm. **c** RT-qPCR analysis of Angpt1 expression in freshly sorted PDGFRβ+ cells from P7 Yap1,Wwtr1iPCKO and control lungs. Data represents mean ± s.e.m. (n = 4 mice, two-tailed unpaired t-test). **d** Angpt1 expression in cultured Verteporfin (VP)-treated (48 h) and control pericytes. Data represents mean ± s.e.m. (n = 4, Welch's t-test). **e** Expression of the indicated transcripts in freshly sorted CD31+ cells from P7 Yap1,Wwtr1iPCKO and control lungs. Data represents mean ± s.e.m. (n = 4 mice, NS not significant, two-tailed unpaired t-test). **f–h** Western blot analysis of Angpt1 protein (**f**; n = 2 controls and 4 mutant mice) and of total and phospho-Tie2 (pTie2) in P12 Yap1,Wwtr1iPCKO and control total lung lysates (**g**, n = 3 controls and 5 mutants). Molecular weight marker (kDa) is indicated. Relative quantification of signals is shown in **h**. Two-tailed unpaired t-test. **i** Scheme showing the time points of tamoxifen administration and analysis for Angpt1 iPCKO mice. **j**, **k** 3D reconstruction confocal images of P12 Angpt1iPCKO and littermate control lungs stained for AQP5 (green), PDGFRβ (red), and PECAM1 (blue). Panels in **k** show higher magnification of PECAM1 staining. Scale bar, 50 μm (**j**) and 30 μm (**k**). **l** Quantitation of airspace volume in P12 Angpt1iPCKO and littermate control lung sections with 3D reconstruction surface images. Data represents mean ± s.e.m. (n = 4 mice; p < 0.0001, two-tailed unpaired t-test). **m** 3D reconstruction confocal images of P12 Angpt1iPCKO and littermate control lungs stained for αSMA (red) and tropoelastin (blue). Scale bar, 50 μm. **n** Quantitation of staining intensity for αSMA or tropoelastin shown in **m**. Intensity was normalized to the size of the parenchymal region. Data represents mean ± s.e.m. (n = 4 mice; NS not significant, two-tailed unpaired t-test). **o** Schematic summary of findings. Pulmonary pericytes regulate ECs and alveolar epithelial cells via angiocrine factors such as angiopoietin-1 and HGF

transcription start site of *Angpt1* gene. DNA fragments encompassing approximately 500 bp of candidate binding regions carrying BamHI and SalI restriction sites were cloned into pGL4 vector downstream of the Luciferase gene with ubiquitin minimal promoter. Mutations in TEAD motif were introduced by site-directed mutagenesis. Cells were co-transfected with the respective reporter constructs and pGL4.75[hRluc/CMV] vector by using Lipofectamine 3000 reagent (Thermo Fisher). Firefly and Renilla luminescence signals were measured 24 h after transfection using Dual-Luciferase reporter assay system (E1910, Promega). Firefly luminescence signals were normalized according to their corresponding Renilla signals resulting in relative luciferase activity. For Verteporfin treatment, pericytes were treated with 0.1, 0.5 or 1 μM Verteporfin (5305, TOCRIS) or DMSO as a vehicle under the protection from light. After 48 h, supernatants were collected for measuring HGF concentration using a quantitative enzyme-linked immunosorbent assay (ELISA) kit (SEA047Mu, Cloud-Clone Corp.). For gene expression analysis, cells were harvested and processed for RNA isolation as described in the "quantitative RT-PCR analysis" section of Methods. For the evaluation of cell toxicity, Annexin V Apoptosis Detection Kit FITC (88-8005-72, Thermo Fisher) and DAPI staining was used following the manufacturer's instructions. After staining, cells were acquired on FACS Aria IIu, followed by the analysis with the FlowJo software. The fractions of Annexin V+ and DAPI−, Annexin V+ and DAPI+ and Annexin V− and DAPI+ were combined as apoptotic cells.

**Generation of polyacrylamide hydrogels and immunofluorescence**. Polyacrylamide hydrogels were prepared according to a previously described protocol[59]. Solutions with varying concentrations of acrylamide (1610140, Bio-Rad)/bis-acrylamide (1610143, Bio-Rad) (3%/0.06 and 8%/0.264%) were mixed to yield hydrogels of different stiffnesses (0.48 and 19.66 kPa, respectively). To allow for cell attachment, fibronectin (356009, Corning) was conjugated to the hydrogel surface using the heterobifunctional linker sulfo-SANPAH. A 1 mg/ml solution of sulfo-SANPAH (803332, Sigma) in milli-Q water was added to the hydrogel surface, and gels were irradiated with 365 nm UV light (intensity of 20 mW/cm$^2$) for 2 min. Substrates were washed with PBS and incubated with a 10 μg/mL fibronectin solution in PBS for 2 h at 37 °C. Samples were washed with PBS prior to cell seeding. Pericytes were plated in growth medium and harvested after 24 h. Cells were fixed with 4%PFA/PBS for 10 min at RT followed by permeabilization with 0.2% Triton X-100 for 10 min at RT. After washing with PBS, the cells were blocked with 1% BSA (A4378, Sigma) for 30 min at RT and incubated with mouse anti-YAP1/TAZ antibody (sc-101199, Santa Cruz, 1:100) in blocking solution over night at 4 °C. After washing with PBS, cells were incubated with donkey anti-mouse IgG AF488 (A21202, Thermo Fisher, 1:200) and AF546 Phalloidin (A22283, Thermo Fisher, 1:100), followed by nuclear counterstaining with DAPI. Z-stacks of confocal images were recorded by LSM880. For quantification of YAP1/TAZ subcellular localization, the middle single optical plane was chosen from Z-stacks of images. The average fluorescence intensity within the region of interest in the nucleus and outside the nucleus (cytoplasm) was measured to determine the nuclear/cytoplasmic ratio. Nuclear/cytoplasmic ratio > 1 was defined as nuclear localization, and nuclear/cytoplasmic ratio < 1 as cytoplasmic localization. A minimum of 50 cells per replicate were examined for each experimental condition.

**RiboTag analysis and RNA sequencing and data analysis**. Immunoprecipitation and purification of ribosome-associated RNA was performed as described previously[23,60] with modifications. Briefly, mouse lungs, brains, hearts, retinas, and inguinal white adipose tissues were perfused through the right ventricle with cold HBSS containing 100 μg/ml cyclohexamide (01810, Sigma) and dissected before snap freezing in liquid nitrogen. Tissues were homogenized in polysome buffer (50 mM Tris-HCl pH 7.4, 100 mM KCl, 12 mM MgCl$_2$, cOmplete ULTRA (Roche), 1% NP-40, 1 mM DTT, 1000 U/ml RNase inhibitor (M0314, New England BioLabs), 200 U/ml SUPERasein (AM2696, Thermo Fisher), 100 μg/ml cyclohexamide, 1 mg/ml heparin) with Pestle (ARgos) and clarified by centrifugation at 20,000 × *g* for 10 min at 4 °C. Before immunoprecipitation, 5% of supernatants were kept as input and the remaining supernatants were mixed with anti-HA-tag Magnetic Beads (M180-11, MBL) and rotated for 8 h at 4 °C. Magnetic beads were washed three times for 5 min in high salt buffer (50 mM Tris-HCl pH 7.4, 300 mM KCl, 12 mM MgCl$_2$, 1% NP-40, 1 mM DTT, 100 μg/ml cyclohexamide). QIAGEN RLT plus buffer was added to magnetic beads and total RNA was isolated using the RNeasy Plus Micro Kit (QIAGEN) according to the manufacturer's instructions. RNA quality was assessed using a 2100 BioAnalyzer (Agilent). Hundred nanogram of RNA were used for preparation of sequencing libraries with the TruSeq Stranded Total RNA Library Prep Kit (Illumina) according to the manufacturer's instructions. Libraries were validated with the BioAnalyzer and quantified by qPCR and Qubit Fluorometric Quantitation (Thermo Fisher). The NextSeq 500/550 High Output v2 Kit or MiSeq Reagent Kit v3 (Illumina) was used for sequencing with a NextSeq 500 or MiSeq (Illumina) to generate 75-bp pair-end reads. Experiments were performed in triplicate.

RNA-seq data analysis was performed as described previously[61] with some modifications. The quality assessment of raw sequence data was performed using FastQC (Version: FastQC 0.11.3). Paired-end sequence reads were mapped to the mm10 mouse genome assembly (GRCm38) using TopHat-2 (Version: tophat-2.0.13)[62]. The mouse genome was downloaded from the iGenome portal. HTSeq was used to count the aligned reads on a per gene basis (Version: HTSeq-0.6.1)[63].

The count data were normalized using the Variance Stabilizing Transformation (VST) function from the DESeq2 package[64]. Principal Component Analysis (PCA) was performed on transformed read counts using the top 500 most variable genes to assess the overall similarity between the samples. Differential gene expression analysis between control and mutants was performed using DESeq2. Differentially expressed genes were selected using a FDR-adjusted p-value cut-off < 0.01 and an absolute log$_2$ fold change > 0.5. Gene symbols were annotated using biomaRt (BioConductor version 3.1). Gene ontology analysis was performed with the DAVID Bioinformatics Resource (v6.7). Heat maps were generated with Morpheus (Broad Institute) or the function heatmap.2 from the R package gplots.

**Western blot analysis**. Mouse lungs were perfused through the right ventricle with cold PBS containing PhosSTOP (Roche) and dissected before snap freezing in liquid nitrogen. Lungs were homogenized in lysis buffer (20 mM Tris-HCl pH 7.4, 1 mM EDTA, cOmplete ULTRA (Roche), PhosSTOP (Roche), 1% NP-40, 0.1% SDS, 150 mM NaCl) with Pestle (ARgos) and clarified by centrifugation at 20000 × *g* for 20 min at 4 °C. Total protein concentration in lysates was quantified using Precision Red Advanced protein assay reagent (Cytoskeleton). Soluble supernatants were prepared in SDS-PAGE sample buffer and analysed by SDS-polyacrylamide gel electrophoresis and immunoblotting after loading 10–30 μg of total lung lysates. Signals were detected with horseradish peroxidase-conjugated secondary antibodies followed by ECL Prime or ECL detection reagent (GE Healthcare). The blots were quantified with the gel analysis function in Fiji software[65]. The following antibodies were used for immunoblotting: mouse anti-β-actin (sc-47778, Santa Cruz, 1:2000), mouse anti-αSMA (A2547, Sigma, 1:1000), rabbit anti-Tropoelastin (ab21600, abcam, 1:1000), rabbit anti-GAPDH (#2118, Cell Signaling, 1:1000), rabbit anti-Phospho-HGF R/c-Met (Y1234/Y1235) (AF2480, R&D Systems, 1:500), mouse anti-Met (#3127, Cell Signaling, 1:1000), mouse anti-Angiopoietin-1 (MAB923, R&D Systems, 1:500), rabbit anti-Phospho-Tie-2 (Y992) (AF2720, R&D Systems, 1:500), goat anti-Tie-2 (AF762, R&D Systems, 1:2000), rabbit anti-YAP (#14074, Cell Signaling, 1:1000), rabbit anti-TAZ (#4883, Cell Signaling, 1:1000), donkey anti-Rabbit IgG, HRP-linked whole Ab (NA934, GE-Healthcare, 1:15000), sheep anti-Mouse IgG, HRP-linked whole Ab (NA931, HG-Healthcare, 1:10000) and Peroxidase AffiniPure Bovine anti-Goat IgG (H + L) (805-035-180, Jackson ImmunoResearch, 1:15000). Full blots are shown in Supplementary Fig. 9.

**Quantitative RT-PCR analysis**. Total RNA was isolated from flow cytometrically sorted or cultured cells using the RNeasy Plus Micro Kit (QIAGEN). 100 ng of RNA per reaction for freshly sorted or 500 ng for cultured cells was used to generate complementary DNA with the iScript cDNA Synthesis Kit (BIO-RAD). The quantitative PCR (qPCR) was performed on a CFX96 Touch Real-Time PCR Detection System (BIO-RAD). The following FAM-conjugated gene expression probes (ThermoFisher) were used in combination with SsoAdvanced Universal Probes Supermix (BIO-RAD): *Pdgfrb* (Mm00435546_m1), *Pecam1* (Mm01242584_m1), *Hgf* (Mm01135184_m1), *Met* (Mm01156972_m1), *Wnt11* (Mm00437327_g1), *Tgfb2* (Mm00436955_m1), *Bmp4* (Mm00432087_m1), *Angpt1* (Mm00456503_m1), *Tie1* (Mm00441786_m1), *Tek* (Mm00443254_m1), *Angpt2* (Mm00545822_m1), *Cspg4* (Mm00507257_m1), *Notch3* (Mm01345646_m1), *Cldn5* (Mm00727012_s1). VIC-conjugated *Actb* (4352341E) TaqMan probe was used to normalize gene expression. The following primer pairs were used in combination with PowerUp SYBR Green Master Mix (ThermoFisher): *Sftpc* forward (5′-ATGGACATGAGTGACAAAGAGGT-3′), *Sftpc* reverse (5′-CACGAT-GAGAAGGCGTTTGAG-3′), *Actb* forward (5′-ACTGCCGCATCCT CTTCCTC-3′), *Actb* reverse (5′-CCGCTCGTTGCCAATAGTGA-3′). *Actb* was used to normalize gene expression. Per group at least four mice from two independent litters with triplicate reactions for each gene were analyzed to obtain the relative expression differences using the 2$^{-\Delta\Delta Ct}$ method.

**Statistical analysis**. Statistical analysis was performed using Graphpad Prism software or the R statistical environment (http://r-project.org). All data are presented as mean ± s.e.m. unless indicated otherwise. Data were tested for normal distribution and unpaired two tailed student t-tests (if two samples have equal variances) or Welch's *t*-tests (if two samples have unequal variances) were used to determine statistical significance between two groups. For analysis of the statistical significance of differences between more than two groups, we performed one-way ANOVA with Tukey's multiple comparison tests to assess statistical significance with a 95% confidence interval. $P < 0.05$ was considered significant unless stated otherwise. No statistical methods were used to predetermine sample size, which was, instead, chosen according to previous experiments. All experiments were performed independently at least three times unless noted otherwise in the respective figure legends. If lungs were not properly inflated, those mice were excluded from analysis. The experiments were not randomized and the investigators were not blinded to allocation during experiments and outcome assessment.

**Data availability**. All relevant data supporting the results of the present study are included within the article, its Supplementary Information files and can be obtained from the corresponding authors upon reasonable request.

RNA sequencing data is available from the Gene Expression Omnibus (GEO) database: GSE96748.

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

## Acknowledgements

We thank G. Luxán for assistance with figure preparation, K.K. Sivaraj for discussion, H.-W. Jeong for RNA-seq experiments, S. Volkery for confocal microscope with Airyscan detector, and A. Witten and the core facility genomics of the medical faculty of the Westfälische Wilhelms-University of Muenster. Prof. Yoshikazu Nakaoka (National Cerebral and Cardiovascular Center Research Institute) for providing Angpt1lox/lox mice, Prof. Sean J. Morrison (University of Texas Southwestern Medical Center) for providing Angpt1tm1.1Sjm mouse and Kenjiro Adachi (Max Planck Institute for Molecular Biomedicine) for providing pGL4 vectors. The Max Planck Society, the University of Muenster, the Deutsche Forschungsgemeinschaft cluster of excellence 'Cells in Motion' (R.H.A.), the Institute for Basic Science (IBS-R025-D1-2015 to G.Y.K.) funded by the Ministry of Science and ICT, Republic of Korea, the EMBO Long-Term Fellowship program (K.K.) and the Uehara Memorial Foundation (K.K.) have supported this study.

## Author contributions

K.K. and R.H.A. designed the study, interpreted the results and wrote the manuscript. R.D.H. characterized the mutant mouse lines. K.K. directed S.K.A. and carried out the immunohistochemistry, immunoblots, qPCR, confocal imaging and quantifications. S.A. generated Pdgfrb(BAC)-CreERT2 and Cdh5-mT/nG mice. M.S. conducted flow cytometry experiments. B.T. prepared hydrogels for cell culture. D.Y.P., S.P.H., and G.Y.K. provided Angpt1 mutant, Angpt1GFP mice samples and COMP-Ang1, J.L.W. Yap1lox/lox and Wwtr1lox/lox mice.
