## [Peer Review File · Nature Communications]

Reviewer #1 (Remarks to the Author):

Summary of Findings:

This is an excellent study, demonstrating the critical role of pericytes in pulmonary alveologenesis. The authors use confocal microscopy to demonstrate the localization of pericytes and their dramatic changes in morphology during critical periods of mouse lung alveologenesis. They demonstrate the close apposition of pericytes to alveolar epithelial cells and use *Pdgfrb(BAC)CreT* mice to mark the development of the pericytes. Their work convincingly demonstrates the specificity of the transgene in the pericytes and carefully assesses changes in the pericyte morphology during postnatal lung development. They used the *Pdgfrb(BAC)CreT* transgenic mouse to specifically delete YAP and TAZ together in pericytes, resulting in inhibition of alveolarization, demonstrating both the critical and novel role for the YAP/TAZ and that of pericytes per se in lung morphogenesis. RNA-Seq was used to identify potential paracrine interactions that contribute to the alveolar defects; they demonstrate that HGF signaling was dependent upon YAP. HGF activated c-Met in alveolar Type 2 cells in vivo and in vitro; thus, YAP regulates HGF to influence epithelial proliferation. Similarly, loss of YAP inhibited angiopoietin-1 (*Angpt1*) production in pericytes in vivo and in vitro; and demonstrated that deletion of *Angpt1* in pericytes caused defects in alveologenesis that were similar to YAP/TAZ deletion. They demonstrate that pericyte YAP nuclear localization was dependent on substrate stiffness in vitro, supporting the concept that pericytes respond to stiffness during development to direct paracrine signaling to the epithelium, in turn, influencing alveologenesis.

Overview: This is an outstanding manuscript, providing new insights into pericyte function and the role of the YAP/TAZ in lung morphogenesis. Novel data regarding paracrine signaling via HGF/*Angpt1* between pericytes and the developing epithelium is shown. The imaging is outstanding and convincing. The field will be most interested in both the morphological assessment of pericyte development and the new role of pericytes in a paracrine network critical for alveolarization. While roles of endothelium and fibroblasts are well-described during lung morphogenesis (for example, *Pdgfra*⁺ fibroblasts), this is the first work to strongly support the role of pericytes. The work is carefully written, well referenced, the methods are clear, and data are appropriate, and analyzed with appropriate statistics.

Comments:

1. The authors might comment and/or provide supplementary data regarding the single TAZ and single YAP1 deletants. How do the authors explain the requirement for deletion of both? Note that deletion of TAZ (complete TAZ^{-/-} KO) is known to cause alveolar simplification (PMID: 18172001, Makita et al.).
2. The authors provide data regarding cell proliferation, it would be important to understand whether there also were changes in apoptosis or cell death in the double mutants. The genotypes and TAM exposure to "controls" (described in the Figures and Methods) should be made clear;

genotypes and treatments of "controls" should be better documented since TAM can alter maturation, growth, and development in the postnatal period. There is potential toxicity from TAM, as well as toxicity from Cre, that should be addressed. Are there alterations in somatic growth of the pups (decreased somatic growth will influence septation)? Was growth or other organ development altered by TAM or CreT with or without the floxed alleles? Since recombination is likely to occur in other tissues, does this occur and influence lung growth or growth of other organs?

3. Verteporfin is known to be remarkably light sensitive and potentially toxic. How were the in vitro cell cultures handled? Was there evidence of cell toxicity and was a dose response performed? (PMID: 28429726, Konstantinou et al.)

4. Since the general deletion of TAZ^{-/-} mice results in airspace abnormalities similar to the present findings, are findings in the TAZ^{-/-} mice consistent with the present results obtained from pericyte deletion? Is the HGF/c-Met/Ang network also altered in the TAZ^{-/-} mice? (Note PMID: 18172001, Makita et al.)

Reviewer #2 (Remarks to the Author):

This is a well written manuscript. The experiments are well thought through, and the MET/HGF axis involvement is quite novel.

It would be useful to further discuss the role of MET/HGF in potential disease states that could arise.

As a minor comment, please be consistent with MET, or c-Met, or c-MET throughout the manuscript.

Reviewer #3 (Remarks to the Author):

In their report, the authors present data that identifies pericytes as regulators of epithelial and endothelial cells during postnatal lung morphogenesis. Mice deleted for the Hippo pathway transcriptional regulators YAP1 and TAZ in pericytes, showed defective alveologenesis. The gene deletions did not affect pericyte numbers, but altered gene expression profiles of pericytes, including decreased expression of hepatocyte growth factor (Hgf) that led to impaired activation of c-Met on alveolar epithelial cells. Furthermore, pericyte deletions of YAP1 and TAZ decreased Angpt1 expression by pulmonary pericytes, which was suggested to decrease Tie2 signalling in

pulmonary endothelial cells in a paracrine manner. Lack of these signalling circuits correlated with decreased endothelial and epithelial cell proliferation. The results thus reveal novel mechanisms of lung morphogenesis mediated via YAP/TAZ-induced paracrine signalling between pulmonary pericytes and endothelial and alveolar epithelial cells in postnatal lung.

Major comments:

1. The functional severity of the defective lung morphogenesis in mutant mice remains unclear - does the emphysema-like phenotype of *Yap1,Wwtr1iPCKO* mice affect physiology in basal or challenged conditions? It would be informative to provide images from HE stained lung sections of *Yap1,Wwtr1iPCKO* and of *Angpt1* knockout mice.
2. The function of *Angpt1* in lung morphogenesis is a novel finding. Does the pulmonary phenotype of the *Angpt1* knockout mice persist after P12, in adult mice? *Angpt1* has been reported as dispensable in mice after E12.5, unless challenged with various vascular pathologies (Jeansson et al, 2012, JCI). The authors should comment on this previous finding, in light of their results.
3. If *Angpt1* is the critical signal decreased in *Yap1,Wwtr1iPCKO* mice to cause defective lung morphogenesis, the authors may try to rescue the phenotype by administering recombinant *Angpt1* in the newborn mice.
4. Endothelial cells appear to express varying amounts of *Hgf* when compared to pericytes (Figure 4e and g). Since endothelial *Hgf* has been reported as a key mediator of reciprocal endothelial-epithelial signaling during lung morphogenesis, levels of *Hgf* expression should be analysed in *Angpt1* deleted mice.
5. *Angpt2* is considered an endothelial cell derived factor - is *Angpt2* expressed by pulmonary pericytes as shown in Figure 4c and Supplementary Figure 6b? *Angpt2* expression appears decreased in *Yap1,Wwtr1iPCKO* deleted mice. Since *Angpt2* can in certain vascular beds (e.g. in the lymphatic vasculature) function as a *Tie2* agonist, is it possible that *Angpt2* also affects lung morphogenesis via endothelial *Tie2*?
6. For vascular quantification in Figure 3g, it would be helpful to visualize the total tissue area by including e.g. nuclear or HE staining.
7. In addition to verteporfin treatment, analysis of *Angpt1* (Fig. 5d) and *Hgf* mRNA (Fig. 4i) expression in shYAP/TAZ silenced pericytes would strengthen the results. Is *Angpt1* a transcriptional target of YAP/TAZ co-activator complexes, as suggested for *Angpt2* (Choi et al, 2015, Nat Commun)?
8. Supplementary Figure 6b. What is the difference between the two rightmost gene lists?
9. Supplementary Figure 7e. Pericytes on soft fibronectin-coated hydrogels do not spread, making it difficult to distinguish between cytoplasmic and nuclear YAP/TAZ staining based on the images in Fig. 7e. More details (and preferentially an image demonstrating this) how quantification between nucleus and cytoplasm was performed should be added. An alternative would be to use slightly stiffer hydrogels to allow cell spreading in both soft and stiff hydrogels.

Reviewer #4 (Remarks to the Author):

I was asked by the editor to comment specifically on the RiboTag results.

Since the authors simply refer to the original RiboTag paper for methods, there are missing pieces of information that make it impossible for a reader to evaluate the underlying experimental data. For example: 1) What is the efficiency and reproducibility of the pull-down of the pericyte-specific transcripts? 2) What is the enrichment of the cell-specific transcripts; i.e. what is the degree of contamination by transcripts from other cell types? Knowledge of these parameters is particularly important since the paper is addressing low abundance transcripts (often <50 rpkm). Since the authors state that they took input samples, it should be possible to provide supporting information as a supplement.

I'm not sure exactly why the RiboTag results are even being presented, since it's not clear what unique conclusions they enable the authors' to draw. The reliability of the RiboTag data should be established or the results left out of the paper.

First of all, we would like to thank the reviewers for their time, effort and valuable suggestions, which have allowed us to improve the manuscript and has also led to the inclusion of a substantial amount of new experimental data. Changes are marked in red in the manuscript file and can be therefore easily identified. Below you will find a detailed point-by-point response to all individual questions and comments.

Reviewer #1 (Remarks to the Author):

Summary of Findings:

This is an excellent study, demonstrating the critical role of pericytes in pulmonary alveologenesis. The authors use confocal microscopy to demonstrate the localization of pericytes and their dramatic changes in morphology during critical periods of mouse lung alveologenesis. They demonstrate the close apposition of pericytes to alveolar epithelial cells and use *Pdgfrb(BAC)CreT* mice to mark the development of the pericytes. Their work convincingly demonstrates the specificity of the transgene in the pericytes and carefully assesses changes in the pericyte morphology during postnatal lung development. They used the *Pdgfrb(BAC)CreT* transgenic mouse to specifically delete YAP and TAZ together in pericytes, resulting in inhibition of alveolarization, demonstrating both the critical and novel role for the YAP/TAZ and that of pericytes per se in lung morphogenesis. RNA-Seq was used to identify potential paracrine interactions that contribute to the alveolar defects; they demonstrate that HGF signaling was dependent upon YAP. HGF activated c-Met in alveolar Type 2 cells in vivo and in vitro; thus, YAP regulates HGF to influence epithelial proliferation. Similarly, loss of YAP inhibited angiopoietin-1 (*Angpt1*) production in pericytes in vivo and in vitro; and demonstrated that deletion of *Angpt1* in pericytes caused defects in alveologenesis that were similar to YAP/TAZ deletion. They demonstrate that pericyte YAP nuclear localization was dependent on substrate stiffness in vitro, supporting the concept that pericytes respond to stiffness during development to direct paracrine signaling to the epithelium, in turn, influencing alveologenesis.

Overview: This is an outstanding manuscript, providing new insights into pericyte function and the role of the YAP/TAZ in lung morphogenesis. Novel data regarding paracrine signaling via HGF/*Angpt1* between pericytes and the developing epithelium is shown. The imaging is outstanding and convincing. The field will be most interested in both the morphological assessment of pericyte development and the new role of pericytes in a paracrine network critical for alveolarization. While roles of endothelium and fibroblasts are well-described during lung morphogenesis (for example, *Pdgfra+* fibroblasts), this is the first work to strongly support the role of pericytes. The work is carefully written, well referenced, the methods are clear, and data are

appropriate, and analyzed with appropriate statistics.

Reply: We appreciate these positive comments highlighting the novelty and importance of our work.

Reviewer #1: 1. The authors might comment and/or provide supplementary data regarding the single TAZ and single YAP1 deletants. How do the authors explain the requirement for deletion of both? Note that deletion of TAZ (complete TAZ^{-/-} KO) is known to cause alveolar simplification (PMID: 18172001, Makita et al.).

Reply: We agree. Previous work (Makita et al, 2008; Mitani et al, 2009) has made use of global and constitutive *Taz/Wwtr1* knockout mice so that the observed phenotypic alterations might reflect changes in alveolar epithelial cells, endothelial cells, fibroblasts, and multiple other relevant cell types. Accordingly, as Supplementary Fig. 4c, d shows, TAZ immunostaining can be prominently seen in non-vascular cells. Importantly, epithelial or endothelial cells are not targeted by *Pdgfrb-CreERT2* (Fig. 11) and TAZ immunosignals are still visible in *Wwtr1*^{iPCKO} lung sections with the exception of pericytes (Supplementary Fig. 4d). Furthermore, our analysis of *Pdgfrb-CreERT2*-induced *Yap1* or *Wwtr1* single mutants (Supplementary Fig. 4e, f) shows that these mice do not display overt lung defects as those seen in double mutants, arguing for functional redundancy of the two gene products in pericytes.

Reviewer #1: 2. The authors provide data regarding cell proliferation, it would be important to understand whether there also were changes in apoptosis or cell death in the double mutants. The genotypes and TAM exposure to "controls" (described in the Figures and Methods) should be made clear; genotypes and treatments of "controls" should be better documented since TAM can alter maturation, growth, and development in the postnatal period. There is potential toxicity from TAM, as well as toxicity from Cre, that should be addressed. Are there alterations in somatic growth of the pups (decreased somatic growth will influence septation)? Was growth or other organ development altered by TAM or CreT with or without the floxed alleles? Since recombination is likely to occur in other tissues, does this occur and influence lung growth or growth of other organs?

Reply: Following the reviewer's suggestions, we have analyzed the number of apoptotic cells in the double mutants using activated caspase 3 staining as a marker. It turns out that there is no significant difference between controls and mutants (Supplementary Fig. 5d).

As requested by the reviewer, we have also clarified our description of ‘control’ mice (see Methods). Controls are littermates of the mutants and both groups have been simultaneously treated with tamoxifen so that potentially toxic, phenotype-altering effects of the drug can be excluded. The influence of Cre can be also excluded because there are obvious differences between double heterozygous mice harboring Cre (*Yap1* +/-, *Wwtr1* +/-, *Cre*+) and double KO mutants (*Yap1* -/-, *Wwtr1* -/-, *Cre*+) . As discussed above, *Pdgfrb-CreERT2*-induced *Yap1* or *Wwtr1* and therefore *Cre*+ single mutants also display no overt morphological alterations of the lung.

There are differences in body weight between *Yap1,Wwtr1*^{iPCKO} double mutants and littermate controls, which are low at P7, a stage where alterations in double mutant lungs are already overt, and increase gradually at P12 and P18 (Supplementary Fig. 5a). The ratio of lung volume and body weight is comparable between controls and mutants, whereas the average airspace volume per average lung volume is much higher in the mutants already at P7 (see below).

Together, these data indicate that retarded growth is the consequence rather than the cause of lung defects and that defective alveologenesis is not merely reflecting the reduced body weight of *Yap1,Wwtr1*^{iPCKO} double mutants. These animals show focal hemorrhaging in the brain, which is very mild at P7 and therefore unlikely to affect lung morphogenesis.

Reviewer #1: 3. Verteporfin is known to be remarkably light sensitive and potentially toxic. How were the in vitro cell cultures handled? Was there evidence of cell toxicity and was a dose response performed? (PMID: 28429726, Konstantinou et al.)

Reply: Agree. As the reviewer pointed out, Verteporfin is light sensitive and these experiments were performed under light protection (which is now mentioned in Methods). We have addressed potential cell toxicity caused by Verteporfin with Annexin V and DAPI staining, which reflect early and late apoptotic cells, respectively. We could not detect the evidence of cell toxicity in a dose dependent fashion under our experimental conditions (Supplementary Fig. 6d). Moreover, we have

confirmed that the expression of *Angpt1* and *Hgf* is suppressed by Verteporfin in a dose-dependent fashion (Supplementary Fig. 6e and 7a).

Reviewer #1: 4. Since the general deletion of TAZ^{-/-} mice results in airspace abnormalities similar to the present findings, are findings in the TAZ^{-/-} mice consistent with the present results obtained from pericyte deletion? Is the HGF/c-Met/Ang network also altered in the TAZ^{-/-} mice? (Note PMID: 18172001, Makita et al.)

Reply: As mentioned above, global deletion of *Wwtr1* is likely to reflect their function in many different cell types including epithelial cells, endothelial cells, pericytes, and fibroblasts. Moreover, the global KO is constitutive and therefore affecting processes from the earliest stages of embryogenesis, whereas we triggered gene inactivation postnatally by administering tamoxifen. This may also explain why airspace abnormalities in global TAZ^{-/-} mice (Makita et al, 2008; Mitani et al, 2009) appear more severe than those reported in our manuscript. Another important difference is that our approach did not significantly alter the number of pericytes in *Yap1, Wwtr1*^{iPCKO} double mutants, which means that our approach has uncoupled the role of the two transcriptional co-regulators in growth control from paracrine signaling by pericytes. Previously published microarray analysis (Mitani et al, 2009) did not show alternations in HGF/c-Met and Angiopoietin/Tie2 signaling, which can be caused by multiple factors: First, the microarray analysis was performed at embryonic day 15.5 and alterations in the two signaling pathways might not occur during early stages of lung development. Second, alterations in HGF/c-Met and Angiopoietin/Tie2 signaling might be masked by more severe defects in other cell types or pathways.

Reviewer #2 (Remarks to the Author):

This is a well written manuscript. The experiments are well thought through, and the MET/HGF axis involvement is quite novel.

Reply: We are very grateful for the kind words and the positive assessment.

Reviewer #2: It would be useful to further discuss the role of MET/HGF in potential disease states that could arise.

Reply: We agree and have extended the Discussion accordingly:

‘It has been reported that low HGF concentrations in tracheal aspirate fluid samples of human neonates are associated with more severe bronchopulmonary dysplasia (BPD), a common chronic lung disease affecting premature infants⁴¹. Exome-sequencing has also identified nonsynonymous mutations in the human *HGF* gene as a genetic factor predisposing to BPD²⁴. In addition, treatment with recombinant HGF improves lung defects in the hyperoxia-induced neonatal mice model of BPD⁴². Mice lacking c-Met in AT2 epithelial cells show an increase in oxidative stress, which is relevant to chronic obstructive pulmonary disease (COPD)^{26,43}. Reduced HGF secretion in peripheral lung of COPD patients correlates with disease severity⁴⁴. Future work should therefore consider pulmonary pericytes in the context of pathologically altered HGF production.’

Reviewer #2: As a minor comment, please be consistent with MET, or c-Met, or c-MET throughout the manuscript.

Reply: Agree. Following the reviewer’s suggestion, we now use ‘c-Met’ throughout the manuscript.

Reviewer #3 (Remarks to the Author):

In their report, the authors present data that identifies pericytes as regulators of epithelial and endothelial cells during postnatal lung morphogenesis. Mice deleted for the Hippo pathway transcriptional regulators YAP1 and TAZ in pericytes, showed defective alveologenesis. The gene deletions did not affect pericyte numbers, but altered gene expression profiles of pericytes, including decreased expression of hepatocyte growth factor (Hgf) that led to impaired activation of c-Met on alveolar epithelial cells. Furthermore, pericyte deletions of YAP1 and TAZ decreased *Angpt1* expression by pulmonary pericytes, which was suggested to decrease Tie2 signalling in pulmonary endothelial cells in a paracrine manner. Lack of these signalling circuits correlated with decreased endothelial and epithelial cell proliferation. The results thus reveal novel mechanisms of lung morphogenesis mediated via YAP/TAZ-induced paracrine signalling between pulmonary pericytes and endothelial and alveolar epithelial cells in postnatal lung.

Reply: We are grateful for this summary of our study.

Major comments:

Reviewer #3: 1. The functional severity of the defective lung morphogenesis in mutant mice remains unclear - does the emphysema-like phenotype of *Yap1,Wwtr1*^{iPCKO} mice affect physiology in basal or challenged conditions? It would be informative to provide images from HE stained lung sections of *Yap1,Wwtr1*^{iPCKO} and of *Angpt1* knockout mice.

Reply: In response to the reviewer's suggestion, we have investigated the measurement of peripheral capillary oxygen saturation (SpO₂) and respiratory rate in basal conditions, which revealed no significant difference between controls and mutants (Supplementary Fig. 5b). To show the emphysema-like phenotype more clearly, we have also included single optical section images of Aquaporin 5 and DAPI stained *Yap1 Wwtr1* ^{iPCKO} and *Angpt1* ^{iPCKO} mice (Supplementary Fig. 4i and 7d). These optical sections show the morphological alterations quite clearly and we have therefore not added H&E staining data because this would be redundant with the images mentioned above.

Reviewer #3: 2. The function of *Angpt1* in lung morphogenesis is a novel finding. Does the pulmonary phenotype of the *Angpt1* knockout mice persist after P12, in adult mice? *Angpt1* has been reported as dispensable in mice after E12.5, unless challenged with various vascular pathologies (Jeansson et al, 2012, JCI). The authors should comment on this previous finding, in light of their results.

Reply: The study of Jeansson et al. shows that *Angpt1* is dispensable for survival after embryonic day (E) 13.5. A recent paper from the same group has shown that Angiopoietin-1 is required for development of the Schlemm's canal in the eye and defects were observed after gene inactivation at E 16.5 (Thomson BR et al., J Clin Invest. 2017). Defects in the Schlemm's canal were also independently reported in mice after postnatal inactivation of the *Angpt1* gene (Kim J et al., J Clin Invest. 2017). Moreover, angiopoietin-1 plays a proangiogenic role in the retina during postnatal development (Lee J et al., Sci Transl Med. 2013). These inconsistencies could be the result of different temporal windows of *Angpt1* inactivation and phenotypic analysis of the resulting mutants, of variations in genetic background and/or of different strategies for the identification of vascular phenotypes. We have, however, not investigated adult *Angpt1* mutants because *Yap1,Wwtr1*^{iPCKO} double mutant mice cannot survive until adulthood so that we felt that this question is beyond the scope of the current manuscript.

Reviewer #3: 3. If *Angpt1* is the critical signal decreased in *Yap1,Wwtr1*^{iPCKO} mice to cause defective lung morphogenesis, the authors may try to rescue the phenotype by administering recombinant *Angpt1* in the newborn mice.

Reply: Agree. We have conducted the rescue experiments with the administration of recombinant COMP-Angiopoietin-1 to *Yap1,Wwtr1*^{iPCKO} mice. We utilized 12.5 $\mu\text{g}/\text{kg}$ of COMP-Angiopoietin-1 every second day from P4 to P10 and analyzed the pups at P12. Morphological analysis revealed an incomplete but statistically significant rescue by COMP-Ang1 relative to PBS-injected mutants (Supplementary Fig. 7h, i).

Reviewer #3: 4. Endothelial cells appear to express varying amounts of *Hgf* when compared to pericytes (Figure 4e and g). Since endothelial *Hgf* has been reported as a key mediator of reciprocal endothelial-epithelial signaling during lung morphogenesis, levels of *Hgf* expression should be analysed in *Angpt1* deleted mice.

Reply: Agree. This suggestion led to some extremely interesting new findings. We have investigated the expression of *Hgf* in sorted pericytes and endothelial cells from *Angpt1* mutant mice by qPCR. This revealed that the expression of *Hgf* in *Angpt1*^{iPCKO} pericytes is significantly reduced (Supplementary Fig. 7e). We further addressed the underlying molecular mechanism and found that treatment with COMP-Ang1 significantly induced the expression of *Hgf* in cultured lung pericytes (Supplementary Fig. 7f), which is consistent with a previous report showing up-regulation of HGF in angiopoietin-1-stimulated cultured smooth muscle cells (Kobayashi H et al., Blood 2006). Next, we addressed the identity of the relevant receptor. Besides *Tie2*, which shows only very low expression in pulmonary pericytes at P7 (Fig. 5a), effects of angiopoietin-1 can be also mediated by $\beta 1$ integrin (Hakanpaa L et al., Nat. Commun. 2014) or by integrin $\alpha v\beta 5$ (Lee J et al., Sci Transl Med. 2013). In line with the latter, treatment with anti-integrin αv blocking antibody completely suppresses the *Angpt1*-triggered increase in *Hgf* expression in cultured pulmonary pericytes (Supplementary Fig. 7g). Together, the new findings substantially extend our mechanistic analysis by showing that angiopoietin-1 acts in an autocrine fashion and regulates *Hgf* levels in pulmonary pericytes. In addition, it is very likely that angiopoietin-1 will also act on pulmonary endothelial cells to coordinate the activity of multiple cell types during alveologenesi. The Discussion and the model in Fig. 5o have been updated accordingly.

Reviewer #3: 5. *Angpt2* is considered an endothelial cell derived factor - is *Angpt2* expressed by

pulmonary pericytes as shown in Figure 4c and Supplementary Figure 6b? *Angpt2* expression appears decreased in *Yap1, Wwtr1*^{IPCKO} deleted mice. Since *Angpt2* can in certain vascular beds (e.g. in the lymphatic vasculature) function as a *Tie2* agonist, is it possible that *Angpt2* also affects lung morphogenesis via endothelial *Tie2*?

Reply: Indeed, qPCR gene expression analysis in sorted pericytes and endothelial cells from P7 *Yap1 Wwtr1*^{IPCKO} mutant mice revealed that the expression of *Angpt2* is higher in pericytes than in endothelial cells and down-regulated in mutants (see below).

Interestingly, a recently published single cell RNA-seq analysis of vascular cells in adult brain and lung also indicates that *Angpt2* is expressed in pericytes of the lung but less in brain (Vanlandewijck M et al., Nature 2018, and data shown below – pericytes (PC) marked in red). This further supports the idea that pericytes acquire organ-specific specialization and molecular properties. The expression level of *Angpt2* is, however, lower than that of *Angpt1* (Figure. 4c). Overall, we felt that a detailed characterization of *Angpt2* requires a separate study and is not within the scope of the current manuscript.

Figure D: [Lung data] Average expression in each cluster

Figure B: [Brain data] Average expression in each cluster

Reviewer #3: 6. For vascular quantification in Figure 3g, it would be helpful to visualize the total tissue area by including e.g. nuclear or HE staining.

Reply: As suggested by reviewer, we have replaced the images including nuclear DAPI staining (Fig. 3g).

Reviewer #3: 7. In addition to verteporfin treatment, analysis of Angpt1 (Fig. 5d) and Hgf mRNA (Fig. 4i) expression in shYAP/TAZ silenced pericytes would strengthen the results. Is Angpt1 a transcriptional target of YAP/TAZ co-activator complexes, as suggested for Angpt2 (Choi et al, Nat Commun 2015,)?

Reply: Following the reviewer's suggestions, we have tried to silence the expression of both *Yap1* and *Wwtr1* in cultured primary pulmonary pericytes with different siRNAs and a lentiviral shRNA system. These attempts were unsuccessful, which might be due to technical reasons or because the prolonged depletion of these two critical proteins is incompatible with the proliferation or survival of our cultured primary cells.

As an alternative approach, we have searched for known TEAD consensus motifs in the genomic sequence of murine *Angpt1* (up to 50 kbp from the transcription start). Several different regions were cloned upstream of a luciferase reporter for dual-luciferase reporter assays. One out of tested elements was able to activate the transcription of a luciferase reporter, whereas point mutations in the TEAD motif reduced reporter gene expression to control levels (Supplementary Fig. 7c). In addition, silencing of all 4 *Tead* genes (*Tead1-4*) significantly reduced the expression of *Angpt1* in pulmonary pericytes (Supplementary Fig. 7b).

Reviewer #3: 8. Supplementary Figure 6b. What is the difference between the two rightmost gene lists?

Reply: We apologize for the confusion. This is one long list that has been subdivided into 3 columns due to space limitations.

Reviewer #3: 9. Supplementary Figure 7e. Pericytes on soft fibronectin-coated hydrogels do not spread, making it difficult to distinguish between cytoplasmic and nuclear YAP/TAZ staining based on the images in Fig. 7e. More details (and preferentially an image demonstrating this) how quantification between nucleus and cytoplasm was performed should be added. An alternative would be to use slightly stiffer hydrogels to allow cell spreading in both soft and stiff hydrogels.

Reply: Agree. Z-stacks of images were acquired, and the middle single optical section was chosen for quantification. The average fluorescence intensity within the region of interest in the nucleus and outside the nucleus (cytoplasm) was measured to determine the nuclear/cytoplasmic ratio. We have defined nuclear/cytoplasmic ratio > 1 as nuclear localization, and nuclear/cytoplasmic ratio < 1 as cytoplasmic localization. We have changed the description in the Methods and have also included single optical section images (Supplementary Fig. 8e).

Reviewer #4 (Remarks to the Author):

I was asked by the editor to comment specifically on the RiboTag results.

Since the authors simply refer to the original RiboTag paper for methods, there are missing pieces of information that make it impossible for a reader to evaluate the underlying experimental data. For example: 1) What is the efficiency and reproducibility of the pull-down of the pericyte-specific transcripts? 2) What is the enrichment of the cell-specific transcripts; i.e. what is the degree of contamination by transcripts from other cell types? Knowledge of these parameters is particularly important since the paper is addressing low abundance transcripts (often <50 rpkm). Since the authors state that they took input samples, it should be possible to provide supporting information as a supplement.

I'm not sure exactly why the RiboTag results are even being presented, since it's not clear what unique conclusions they enable the authors' to draw. The reliability of the RiboTag data should be established or the results left out of the paper.

Reply: We have shown the efficiency and reproducibility of the RiboTag method in a recent paper from our group (Jeong HW et al., Nat. Commun. 2017). We chose RiboTag because it allows us to get clean expression data from different vascular cell types – namely endothelial cells and pericytes, respectively – that are tightly associated and difficult to separate completely by FACS or other methods. Moreover, it is also highly beneficial that the RiboTag approach avoids long tissue dissociation and cell sorting steps, which are known to upregulate stress-induced genes.

In the current manuscript, fold-enrichment by RiboTag was analyzed by comparing input RNA versus anti-HA immunoprecipitated RNA by qPCR for a number of cell type-specific marker genes: namely *Pdgfrb* and *Cspg4* for pericytes, *Cldn5* and *Tie1* for endothelial cells, and *Sftpc* for alveolar type 2 epithelial cells (Supplemental Fig. 6a). Using the RiboTag method in lung led to a more than 10-fold enrichment of pericyte-specific transcripts relative to input, whereas endothelial cell and epithelial cell-specific transcripts were strongly depleted at the same time. The fold-enrichment of a transcript after immunoprecipitation (IP) is in inverse proportion to the target cell abundance in whole tissue, but not fully consistent with the cellular composition because of differences in IP efficiency. The percentage of pericytes in lung is around 5% based on our analysis. Thus, 10-fold enrichment of pericyte marker genes by the IP indicates that the efficiency of the immunoprecipitation against HA is approximately ~ 50%.

Reviewer Comments:

Reviewer #1 (Remarks to the Author):

The authors have revised an interesting and novel manuscript and were highly responsive to the reviewers' suggestions. New data regarding regulation of Angpt1 by the YAP transcription factor TEAD is provided, further strengthening their observations regarding the role of YAP signaling in pericytes to regulate Angpt1, a known regulator of Tie 2/Tek and critical for lung formation. The studies are carefully performed. The roles of pericyte and lung morphogenesis are poorly understood at present, and this is an addition to the field. The imaging is excellent. The work is novel, providing new data supporting important role for pericytes in lung morphogenesis. Furthermore, they demonstrate an important role for the HIPPO/YAP signaling, identifying targets in the network in which pericyte Angpt1 and HGF interact between pericytes and alveolar epithelial cells. I believe the authors have addressed concerns, and feel that the work is an important addition to our understanding of perinatal lung formation.

Reviewer #3 (Remarks to the Author):

The points I raised in the previous round of review have been satisfactorily addressed.

Reviewer #4 (Remarks to the Author):

I have reviewed the authors' responses to my comments on the original manuscript. Jeong et al. (2017) answers my concerns about the methodology; however instead of simply citing the original RiboTag paper (Sanz et al, 2009), Jeong should also be cited. Fig S6a answers my questions about the enrichment of specific transcripts through immunoprecipitation.